# ICPRL: Acquiring Physical Intuition from Interactive Control

## Abstract

VLMs excel at static perception but falter in interactive reasoning in dynamic physical environments, which demands planning and adaptation to dynamic outcomes. Existing physical reasoning methods often depend on abstract symbolic inputs or lack the ability to learn and adapt from direct, pixel-based visual interaction in novel scenarios. We introduce **ICPRL**[1] (In-Context Physical Reinforcement Learning), a framework inspired by In-Context Reinforcement Learning (ICRL) that empowers VLMs to acquire physical intuition and adapt their policies in-context. Our approach trains a vision-grounded policy model via Group Relative Policy Optimization (GRPO) over diverse multi-episode interaction histories. This enables the agent to adapt strategies by conditioning on past trial-and-error sequences, without requiring any weight updates. This adaptive policy works in concert with a separately trained world model that provides explicit physical reasoning by predicting the results of potential actions. At inference, the policy proposes candidate actions, while the world model predicts outcomes to guide a root-node PUCT search to select the most promising action. Evaluated on the diverse physics-based puzzle-solving tasks in the DeepPHY benchmark, ICPRL demonstrates significant improvements across both its policy-only and world-model-augmented stages. Notably, these gains are retained in unseen physical environments, demonstrating that our framework facilitates genuine in-context acquisition of the environment's physical dynamics from interactive experience.

## 1 Introduction

Despite impressive perception, Vision-Language Models (VLMs) struggle with interactive physical reasoning that requires acting, observing consequences, and revising plans. Existing benchmarks largely test static understanding, leaving a gap in closed-loop, physics-grounded competence (Wang et al., 2023; Agarwal et al., 2025). Prior work either distills policy improvement from symbolic histories or scales meta-RL with weight updates; neither directly equips VLMs to learn physics from pixels through interaction, to adapt to new tasks in a zero-shot setting (Bakhtin et al., 2019; Li et al., 2024; Matthews et al., 2025).

We introduce **ICPRL**, an ICRL (Bauer et al., 2023b) -inspired VLM framework that enables

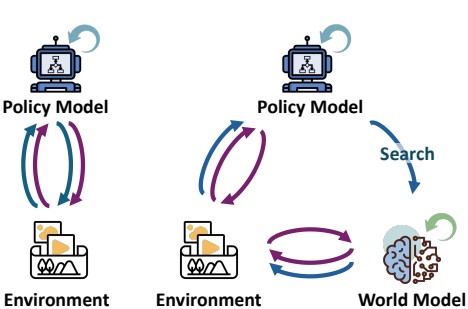

Figure 1: The ICPRL framework integrates world model and adaptive policy.

in-context policy adaptation for interactive physical reasoning. Our approach reinterprets the concept of distilling a policy improvement operator. Instead of training a model to *imitate* the learning trajectory of a separate RL algorithm, our vision-grounded policy model ($\pi_\theta$) directly becomes the subject of policy improvement. We train this VLM using GRPO (Shao et al., 2024) over a vast collection of diverse, multi-episode interaction histories. This process teaches the model *how* to adjust

---

[1]Pronounced IC-Pearl.

its strategy based on historical context, including previous successes and failures. Consequently, the ability to adapt to new unseen task instances is not an emergent phenomenon at test time, but rather a robust capability learned and encoded into the model's weights during training. At inference, $\pi_\theta$ leverages this ability to improve its performance purely by conditioning on recent interaction history within its forward pass, requiring no further gradient updates. Crucially, we augment this adaptive policy with an independently trained world model ($M_\phi$), which acquires and provides explicit physical intuition by predicting action outcomes and dynamics. This world model serves as a powerful in-context planning component, guiding the policy's exploration and enabling the agent to discover task-specific physics and refine its plans more efficiently (Zhang et al., 2025; Chung et al., 2025).

This two-component architecture is illustrated in Figure 1. Our ICPRL agent (**Right**) consists of: (1) an adaptive **Policy Model** that learns to improve from its interaction history, and (2) an independently trained **World Model** that provides physical intuition. Training interactions for our models are denoted by **purple arrows**. At inference time (**blue arrows**), the policy performs an in-context **Search** over plans simulated by the world model, enabling more efficient reasoning. This approach stands in contrast to conventional agents (**Left**), which train a policy to react directly to environmental feedback without an explicit planning module.

We evaluate ICPRL on DeepPHY (Xu et al., 2025), a suite spanning single-shot placement to multi-step control, with standardized annotated observations and structured action spaces that preserve physics while enabling reliable VLM control (Wu et al., 2024). On the complex multi-step *I-PHYRE* task, our full model achieves a 93.3% success rate. More critically, ICPRL shows remarkable generalization to unseen environments; *e.g.,* on the *Pooltool* task, it achieves a 71.0% success rate, more than doubling the performance of strong baselines like GPT-o3. These gains on tasks the model was never trained on demonstrate that our framework facilitates genuine in-context acquisition of the environment's physical dynamics from interactive experience.

Building on this evidence, our contributions are twofold: First, we introduce a novel vision-grounded ICRL-inspired framework for VLMs that adapts at test time without weight updates by using interaction histories to infer physics, guided by an explicit world model. Second, We establish state-of-the-art zero-shot performance on diverse interactive physical reasoning tasks, along with analyses on world-model prediction, action discretization, and history length.

## 2 RELATED WORK

ICRL represents a paradigm shift in how autonomous agents acquire new skills, moving from slow, gradient-based adaptation to rapid ICL within the forward pass of neural networks. Typically, Algorithm Distillation (AD) (Laskin et al., 2023) distills a policy improvement operator into a Transformer by training on numerous learning histories from a source RL algorithm (*e.g.,* A3C (Mnih et al., 2016)). While methods like AD focus on distilling an external RL algorithm—training a Transformer to *imitate* the learning process itself—our **ICPRL** internalizes this capability directly within the policy model. To circumvent high computational cost, AD$^\epsilon$ (Zisman et al., 2024) creates synthetic learning histories from a single demonstrator policy by simulating a learning process with a decaying noise curriculum. An alternative approach is memory-based meta-RL (Bauer et al., 2023a), in which an agent with persistent memory is trained to improve its strategy over repeated trials on the same task. Such techniques have been extended to LLMs by PAPRIKA (Tajwar et al., 2025), fine-tuning on self-generated interaction data using RPO (Pang et al., 2024). SCoRe (Kumar et al., 2025) uses on-policy RL to fine-tune an LLM, teaching it a generalizable ICL skill for multi-turn self-correction. ICAL (Sarch et al., 2024) leverages a VLM to autonomously refine noisy interaction data into high-quality examples, enhancing retrieval-augmented ICL planning. ICRL can also leverage Unsupervised Environment Design (UED) and automated curriculum learning to generate diverse and high-quality training tasks. Building on XLand (Stooke et al., 2021), which advanced the training of generalist agents through an open-ended process of dynamic task generation, Adaptive Agent (AdA) (Bauer et al., 2023a) introduces an automated curriculum method to navigate the immense XLand 2.0 task space. Prioritized Level Replay (PLR) (Jiang et al., 2021b) is a heuristic that focuses training on tasks at the edge of an agent's competence, and has been theoretically formalized and improved (Dennis et al., 2020; Jiang et al., 2021a; Parker-Holder et al., 2022). Recent curriculum strategies have advanced from refining task selection at the learning frontier (Rutherford et al., 2024) to augmenting the training space with generative models (Garcin et al.,

2024), and toward creating truly open-ended domains by evolving the game mechanics themselves (Earle & Togelius, 2024; Frans & Isola, 2023). Instead of mimicking an optimization algorithm, our VLM policy is trained via GRPO (Shao et al., 2024) on multi-episode histories to learn *how to execute* a policy improvement step in-context. By observing diverse histories of trial and error, the VLM learns to condition its future actions on past outcomes, effectively embedding an adaptive, in-context learning skill into its own parameters. This is achieved through gradient-based optimization over multi-episode histories, enabling the VLM to perform in-context adaptation at inference time.

Physical reasoning serves as the foundation for world model construction (Wu et al., 2024; Agarwal et al., 2025) and embodied intelligence tasks (Luo et al., 2023; Yuan et al., 2025). However, most evaluations of LLMs & VLMs focus on static problem-solving benchmarks. These evaluations—often large-scale QA tasks on object properties (Wang et al., 2023; Chow et al., 2025) or text-based physics exams (Wang et al., 2025b; Chung et al., 2025; Zhang et al., 2025)—assess primarily agents' ability to recall scientific knowledge or infer logical outcomes from fixed contexts. While useful for evaluating declarative knowledge, these approaches largely bypass the challenges of real-time visual perception and continuous interaction in dynamic environments. Another line of research investigates physical reasoning in simulated environments but often abstracts away perceptual grounding by relying on symbolic inputs. Common approaches either provide agents with pre-processed symbolic inputs, such as object property matrices (Bakhtin et al., 2019; Li et al., 2024; Matthews et al., 2025), or enable interaction with simulators through code generation (Cherian et al., 2024). While effective for isolating specific planning tasks, these methods limit generalizability by bypassing raw sensory data understanding. Our work addresses this limitation by focusing on interactive physical reasoning, where agents must plan and execute a sequence of actions within a physical simulator, guided by learned intuition. Unlike approaches that employ world models for candidate filtering (Qi et al., 2025) or rely on pre-trained reasoning capabilities (Chen et al., 2025), ICPRL trains the policy to internalize an improvement operator via GRPO, enabling genuine test-time adaptation through a bi-level optimization process where the world model guides a lookahead search. We adopt DeepPHY (Xu et al., 2025) as benchmark, covering 6 diverse dynamic physical environments, to evaluate agents' interactive physical reasoning directly from visual inputs in dynamic environments.

## 3 ICPRL

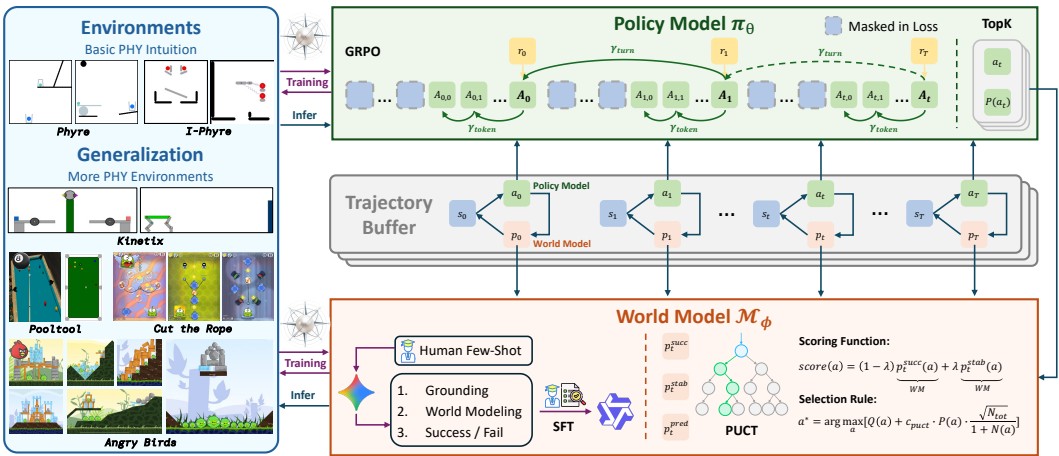

Figure 2: Overview of the **ICPRL** Framework, which decouples policy learning from world modeling for robust in-context planning. **Training Stage:** We separately train a **Policy Model** ($\pi_\theta$) to generate context-aware actions and a **World Model** ($\mathcal{M}_\phi$) to predict physical outcomes. **Inference Stage:** $\pi_\theta$ proposes candidate actions, which $\mathcal{M}_\phi$ evaluates by acting as an *in-context physical simulator*. These evaluations guide a PUCT search to select the optimal action, enabling effective zero-shot planning.

We consider a family of physical simulators $\{\mathcal{E}_m\}_{m=1}^M$. At each decision point we observe $s_t \in \mathcal{S}$, choose an action $a_t \in \mathcal{A}$, and the simulator returns an updated observation and a task reward or

success indicator. Our ICPRL framework comprises distinct training and inference stages, as shown in Figure 2: **Training (two stages):** We foster **in-context adaptive capabilities** within a VLM **policy model** $\pi_\theta$ through online GRPO (Shao et al., 2024) on diverse multi-episode interaction histories. Concurrently, a separate **world model** $\mathcal{M}_\phi$ is trained offline to acquire robust physical intuition by predicting environment dynamics and outcomes. **Inference:** These two models work in concert. $\pi_\theta$ proposes contextually informed actions, and $\mathcal{M}_\phi$, acting as an **in-context physical simulator**, provides crucial outcome predictions to guide a lightweight **root-node PUCT** (Silver et al., 2017) search. This separation keeps policy $\pi_\theta$ learning stable and simulator-grounded, while enabling rapid context-sensitive adaptation and refined planning at test time without further weight updates.

## 3.1 TRAINING STAGE: POLICY MODEL $\pi_\theta$

Our policy model $\pi_\theta$ is a VLM that generates textual outputs in a structured format (detailed in Appendix E). These outputs are parsed into discrete actions $a_t \in \mathcal{A}$ by an environment-specific converter, for interaction with simulators across $M$ environments. The reinforcement signal is derived directly from the task's native rewards (*e.g.,* success/failure).

While standard RL algorithms can be applied to this setup by treating the agent's trajectory as a monolithic sequence, this approach often proves suboptimal. It fails to distinguish between agent-generated tokens (reasoning and actions) and environment-provided context, and it applies a uniform temporal discounting that conflates intra-turn and inter-turn credit assignment. To address these limitations, we enhance our training framework by incorporating principles from selective token masking and bi-level advantage estimation (Wang et al., 2025a). First, we employ **selective token masking** to focus the learning signal; we set $M_t^{\text{loss}} = 1$ for all tokens generated by the policy $\pi_\theta$ and $M_t^{\text{loss}} = 0$ for all other tokens (*e.g.,* those constituting the state observation $s_k$). Second, for multi-turn credit assignment, we adopt a **bi-level advantage estimation** scheme. We set an intra-turn discount $\gamma_{\text{token}} = 1.0$, which treats all tokens within a reasoning-action chain as equally contributing to the turn's outcome. For interactions across turns, we use an inter-turn discount of $\gamma_{\text{turn}} = 0.9$, which allows for standard temporal discounting of long-term rewards. These turn-aware mechanisms are used to compute the final token-level advantage estimates, $A_{i,t}^{\text{GAE}}$, for our GRPO objective.

We sample a group of tokens $G$ generated from policy model $\pi_\theta$, $G = \{o_1, o_2, \ldots, o_G\}$. The importance sampling ratio, denoted as $r_t(\theta)$, is defined as the probability ratio between the new and old policies for a given token: $r_t(\theta) = \pi_\theta(o_{i,t} \mid q, o_{i,<t})/\pi_{\theta_{\text{old}}}(o_{i,t} \mid q, o_{i,<t})$, where $o_{i,t}$ is the $t$-th token of the $i$-th output sample, $o_{i,<t}$ is the context for generating the token $o_{i,t}$. With a frozen reference $\pi_{\text{ref}}$, the masked GRPO losses are:

$$
\mathcal{L}_{\text{GRPO}} = \frac{1}{G} \sum_{i=1}^{G} \min \left( r_t(\theta) \cdot \hat{A}_{i,t}, \ \text{clip}\left(r_t(\theta), 1-\varepsilon, 1+\varepsilon\right) \cdot \hat{A}_{i,t} \right) \tag{1}
$$
$$
- \beta D_{\text{KL}}\left(\pi_\theta \,\|\, \pi_{\text{ref}}\right) - \beta D_{\text{KL}}\left(\pi_\theta(\cdot \mid q, o_{i,<t}) \,\|\, \pi_{\text{ref}}(\cdot \mid q, o_{i,<t})\right)
$$

where the relative advantage $\hat{A}_{i,t}$ for the $i$-th response is $A_i = \frac{r(o_i) - \text{mean}(\{r(o_1),\ldots,r(o_G)\})}{\text{std}(\{r(o_1),\ldots,r(o_G)\})}$, $q$ is input prompt given to $\pi_\theta$, $\varepsilon$ is the PPO clipping hyperparameter, and $\beta$ is the coefficient for the Kullback-Leibler (KL) divergence penalty.

## 3.2 TRAINING STAGE: WORLD MODEL $\mathcal{M}_\phi$

Given a state-action pair $(s, a)$, the world model $\mathcal{M}_\phi$ is trained to return a strictly structured JSON containing a predicted success probability and a natural language prediction.

$$
\mathcal{M}_\phi(s, a) \Rightarrow \begin{cases} \hat{p}_{\text{succ}} \in [0, 1], \\ \hat{p}_{\text{pred}} : \text{NLP prediction} \end{cases} \tag{2}
$$

The predicted success probability $\hat{p}_{\text{succ}}$ is used directly for planning, while $\hat{p}_{\text{pred}}$ provides qualitative insights for analysis.

To enhance planning robustness, a stability score, $\hat{p}_{\text{stab}}$, is derived from the model's own predictions. This score is calculated by evaluating the model's success predictions over a neighborhood of perturbed actions. For an environment-specific action metric $d_{\mathcal{A}}$ and a radius $\delta$, we define the perturbation set as: $\mathcal{B}_{\delta}(a) = \{a' : d_{\mathcal{A}}(a, a') \leq \delta\}$. The stability $\hat{p}_{\text{stab}}$ is the expected model-predicted success rate over actions sampled from this perturbation set. Let $\hat{p}_{\text{succ}}(s, a')$ denote the success probability predicted by $\mathcal{M}_{\phi}(s, a')$. The stability is then defined as:

$$\hat{p}_{\text{stab}}(s, a) = \mathbb{E}_{a' \sim D(\mathcal{B}_{\delta}(a))}[\hat{p}_{\text{succ}}(s, a')]. \tag{3}$$

In practice, this expectation is estimated for each sample by querying the model $\mathcal{M}_{\phi}$ with a small, finite set of "jittered" actions and averaging the resulting $\hat{p}_{\text{succ}}$ predictions.

To train world model $\mathcal{M}_{\phi}$, we first curate a high-quality dataset with a balanced representation of outcomes. This procedure is formalized in Algorithm 1 in Appendix D. The data generation process for each initial state $s$ is as follows:

We first enumerate all possible successful actions from state $s$. If $k$ successful actions are found, we then sample and filter to obtain a corresponding set of $k$ diverse, failing actions. This ensures a balanced 1:1 ratio of positive to negative examples for each state, preventing model bias. Each successful and failing action is executed in the simulator. We record the entire chain reaction following the action, from its completion until the environment reaches a stable state. From this resulting video sequence, we uniformly sample 5 frames to represent the dynamic evolution of the environment post-action. The collected data—comprising the initial state (raw and annotated images), the action, the five dynamic frames, and the ground-truth success/fail label—is used to generate rich training signals. We employ a large VLM with a few-shot prompting strategy to produce the NLP ground truth for $\hat{p}_{\text{pred}}$. The VLM is prompted to generate text describing the objects and their relations in the initial state (**Grounding**), predicting the chain reaction that will occur as a consequence of the action (**World Modeling**), and providing a **Success or Fail Label**. All automatically generated annotations are then manually inspected and corrected by human reviewers to ensure their accuracy and quality.

With this curated dataset, we train world model $\mathcal{M}_{\phi}$. The training objective combines a regression loss for success prediction and a language modeling loss for the textual output. The total loss $\mathcal{L}_{\text{WM}}$ is a weighted sum: $\mathcal{L}_{\text{WM}} = \mathcal{L}_{\text{succ}} + \lambda_{\text{text}}\mathcal{L}_{\text{text}}$, where $\mathcal{L}_{\text{succ}} = \text{BCE}(\hat{p}_{\text{succ}}, y)$ is the Binary Cross-Entropy loss between the predicted success probability and the ground-truth label $y$, and $\mathcal{L}_{\text{text}}$ is a standard cross-entropy loss for training the language prediction component.

### 3.3 Inference Stage

At inference time, for each decision point, we employ a root-node search procedure that integrates a learned policy prior with scores from our world model, $\mathcal{M}_{\phi}$. This process, detailed in Algorithm 2 in Appendix D, unfolds in four stages: candidate generation, scoring, search, and execution.

First, to form a discrete set of candidate actions $A = \{a_i\}_{i=1}^{K}$, we draw $S$ samples from our policy network $\pi_{\theta}(\cdot \mid o)$. Based on these samples, we establish a **frequency prior** $P(a)$ over the candidate set, defined as $P(a) = c(a) / \sum_{a' \in A} c(a')$, where $c(a)$ is the count of action $a$.

Next, each candidate action $a \in A$ is evaluated by $\mathcal{M}_{\phi}$. To obtain robust estimates, we query $\mathcal{M}_{\phi}(o, a)$ for $K$ stochastic forward passes, yielding sets of success and stability predictions; the stability score is derived from environment-specific action perturbations detailed in Appendix E. We compute their means, $\mu_p(o, a)$ and $\mu_s(o, a)$ respectively, and combine them into a single score: $\text{score}(a \mid o) = (1 - \lambda_{\text{PUCT}})\mu_p(o, a) + \lambda_{\text{PUCT}}\mu_s(o, a)$. The standard deviation of these scores across the $K$ passes, denoted by $\sigma$, provides a measure of model uncertainty, which can optionally be used to form an optimistic value estimate, such as $\tilde{Q} = Q + \beta\sigma$.

These scores then guide a search at the root node using PUCT. For a planning budget of $B$ iterations, we maintain visit counts $N(a)$ and mean action values $Q(a)$ for each candidate. In each iteration, we select the action $a_t$ that maximizes the PUCT criterion:

$$a_t = \arg\max_{a \in A}\left[Q(a) + c_{\text{PUCT}} \cdot P(a) \cdot \frac{\sqrt{\sum_{b \in A} N_{tot}(b)}}{1 + N(a)}\right] \tag{4}$$

Upon selecting $a_t$, its world model score is used to update its statistics: $N(a_t)$ is incremented, and $Q(a_t)$ is updated with the new score. To improve efficiency, we employ an **early-stopping** mechanism, terminating the search if the budget $B$ is exhausted, or prematurely if either the same action is chosen for 3 consecutive iterations or an action is found to be highly successful (*i.e.,* $\mu_p(o, a_t) > 0.8$).

Finally, after the search concludes, the action with the highest visit count or mean Q-value, $a^* = \arg\max_a Q(a)$, is selected and executed for a **single** step in the simulator. The resulting transition is logged for evaluation and can be used for future model updates.

## 4 EXPERIMENTS

### 4.1 EXPERIMENTAL SETUP

This subsection details the experimental environments, comparison baselines, evaluation metrics, and the implementation details of ICPRL.

#### 4.1.1 ENVIRONMENTS

We adopt DeepPHY (Xu et al., 2025) as our evaluation platform. This benchmark comprises six diverse and interactive physical simulation environments: *PHYRE* (Bakhtin et al., 2019), *I-PHYRE* (Li et al., 2024), *Kinetix* (Matthews et al., 2025), 🎱 *Pooltool* (Kiefl, 2024), 🐦 *Angry Birds*[2], and 🐸 *Cut the Rope*[3]. Encompassing a wide spectrum of physical properties (*e.g.,* gravity, elasticity, collisions), these environments present challenges ranging from single-step planning (*PHYRE*) to complex multi-step sequential control (*Kinetix*, *Cut the Rope*). This makes the benchmark an ideal platform for testing the generalization and adaptability of our model.

#### 4.1.2 EVALUATION PROTOCOL

To rigorously assess the acquisition of transferable physical intuition, we strictly define our evaluation protocol by partitioning the DeepPHY suite into source and target domains.

We train the Policy Model ($\pi_\theta$) and World Model ($\mathcal{M}_\phi$) exclusively on *PHYRE* and *I-PHYRE*. The models are evaluated on *Kinetix, Pooltool, Angry Birds*, and *Cut the Rope*. We enforce a strict zero-shot inference setting on the Target Domains. No weight updates, fine-tuning, or parameter calibration are permitted on these environments. The agent must rely solely on its learned physical intuition and in-context adaptation capabilities to solve these unseen tasks.

#### 4.1.3 BASELINES

We compare ICPRL with leading closed-source and open-source VLMs, as detailed in Table 6. These closed-source models represent the current SOTA in zero-shot physical reasoning. We use the random action agent (`MOCK`) (provided by DeepPHY) as a lower bound on performance.

We evaluate two distinct variants of our own method: i) **ICPRL (Policy Only):** This variant performs inference using only our policy model, $\pi_\theta$, trained with GRPO, without leveraging the world model or PUCT search. This baseline is designed to measure the performance of the adaptive policy itself. And ii) **ICPRL (Full):** Our complete model, which combines the adaptive policy $\pi_\theta$ with the PUCT search guided by World Model $\mathcal{M}_\phi$, showcasing how both work in tandem.

#### 4.1.4 EVALUATION METRICS

We employ two core metrics for our evaluation: **Success Rate (S.R.)**, which is the percentage of tasks solved successfully; and **Average Attempts** (Avg. Att.), the mean number of attempts taken, calculated only on successfully solved tasks. We report Success Rate and Average Attempts over 3 runs under the different settings.

---

[2]https://apps.apple.com/us/app/rovio-classics-angry-birds/id1596736236
[3]https://apps.apple.com/cn/app/cut-the-rope/id1024507512

### 4.1.5 IMPLEMENTATION DETAILS

**Model Architecture:** Both Policy Model $\pi_\theta$ and World Model $\mathcal{M}_\phi$ are built and trained based on Qwen2.5-VL-3B/7B-Instruct (Qwen Team, 2025).

**Training:** We conducted training for Policy Model $\pi_\theta$ and World Model $\mathcal{M}_\phi$ on the first two environments (*PHYRE* & *I-PHYRE*[4]), and then evaluated their generalization performance on the remaining four. Policy Model $\pi_\theta$ is trained online using GRPO (Shao et al., 2024). For training, we set the actor and critic learning rates to $1 \times 10^{-6}$ and $1 \times 10^{-5}$, respectively. The policy updates are regularized with a KL penalty coefficient ($\beta$) of 0.001. We employ a bi-level advantage estimation with a turn-level discount factor $\gamma_{\text{turn}} = 0.95$ and a token-level discount $\gamma_{\text{token}} = 1.0$. During the online data collection (rollout) phase, actions are sampled using a temperature of 0.7 and top-p nucleus sampling of 0.95. World Model $\mathcal{M}_\phi$ is trained offline on the curated dataset, with the loss weight for the text component, $\lambda_{\text{text}}$, set to 0.2.

**Inference:** The PUCT search is configured with a candidate sample size of $S = 32$, $K = 8$ world model queries per action, a planning budget of $B = 32$, a score mixing weight of $\lambda_{\text{PUCT}} = 0.25$, and an exploration coefficient of $c_{\text{PUCT}} = 1.5$.

### 4.2 MAIN RESULTS

Table 1: Overall Performance on DeepPHY. Model naming conventions are detailed in Appendix B.

| Row Ann. | Policy Model $\pi_\theta$ | World Model $\mathcal{M}_\phi$ | PHYRE Att. 1 | PHYRE Att. 10 | I-PHYRE Att. 1 | I-PHYRE Att. 10 | Kinetix | Pooltool Att. 1 | Pooltool Att. 15 | Angry Birds | Cut the Rope |
|---|---|---|---|---|---|---|---|---|---|---|---|
| # 01. | *MOCK* | - | 1.50% | 10.80% | 23.33% | 53.33% | 21.40% | 2.33% | 48.00% | 17.65% | 11.36% |
| **Open-Source Models** | | | | | | | | | | | |
| # 02. | Qwen-3B | - | 0.17% | 5.85% | 13.33% | 16.67% | 16.22% | 0.00% | 50.00% | 17.65% | 7.95% |
| # 03. | Qwen-7B | - | 0.67% | 10.10% | 32.42% | 32.42% | 13.51% | 23.50% | 26.50% | 20.59% | 9.09% |
| # 04. | Qwen-32B | - | 0.03% | 8.70% | 0.17% | 5.60% | 15.20% | 0.00% | 14.29% | 26.47% | 6.82% |
| # 05. | Qwen-72B | - | 2.43% | 14.92% | 13.33% | 43.33% | 14.86% | 0.00% | 18.00% | 29.41% | 13.64% |
| # 06. | Qwen-72B | Qwen-72B | 1.78% | 10.39% | 10.00% | 36.67% | 12.16% | 0.00% | 14.00% | 26.47% | 11.50% |
| **Close-Source Models** | | | | | | | | | | | |
| # 07. | Claude 4.0 Opus | - | 1.73% | 10.63% | 36.67% | 56.67% | 23.20% | 0.00% | 49.00% | 32.35% | 26.14% 🏅 |
| # 08. | Claude 4.0 Opus | Claude 4.0 Opus | 1.11% | 7.44% | 30.00% | 50.00% | 20.50% | 0.00% | 42.00% | 29.41% | 23.86% 🏅 |
| # 09. | Gemini-2.5-Pro | - | 2.17% | 22.07% | 20.00% | 63.33% | 24.10% | 36.50% | 68.00% | 35.29% | 22.73% |
| # 10. | Gemini-2.5-Pro | Gemini-2.5-Pro | 2.17% | 12.33% | 16.67% | 53.33% | 21.80% | 25.00% | 60.00% | 29.41% | 20.45% |
| # 11. | GPT-o3 | - | 3.03% | 30.77% | 30.00% | 86.67% | 26.89% 🏅 | 0.00% | 25.67% | 35.29% | 18.18% |
| # 12. | GPT-o3 | GPT-o3 | 0.17% | 25.60% | 25.00% | 76.67% | 24.50% 🏅 | 0.00% | 22.00% | 29.41% | 16.50% |
| **Fine-tuned Models** | | | | | | | | | | | |
| **Qwen2.5-VL-3B-Instruct Fine-tuned Series** | | | | | | | | | | | |
| # 13. | Qwen-3B$^{\text{P}}$ $^{SFT}$ | - | 1.33% | 25.58% | 20.00% | 30.00% | 13.70% | 0.00% | 51.00% | 23.53% | 10.23% |
| # 14. | Qwen-3B$^{\text{P}}$ $^{SFT}$ | Qwen-3B$^{\text{P}}$ | 1.52% | 28.69% | 19.87% | 32.45% | 13.55% | 0.00% | 53.67% | 26.47% | 11.85% |
| # 15. | Qwen-3B$^{\text{I}}$ $^{SFT}$ | - | 0.33% | 9.83% | 23.33% | 45.56% | 9.60% | 0.00% | 43.00% | 20.59% | 7.80% |
| # 16. | Qwen-3B$^{\text{I}}$ $^{SFT}$ | Qwen-3B$^{\text{I}}$ | 0.74% | 11.88% | 23.21% | 47.34% | 9.82% | 0.00% | 45.33% | 23.53% | 9.13% |
| # 17. | Qwen-3B$^{\text{P\&I}}$ $^{SFT}$ | - | 10.42% | 29.00% | 20.00% | 50.00% | 15.07% | 0.00% | 57.00% | 26.47% | 11.60% |
| # 18. | Qwen-3B$^{\text{P\&I}}$ $^{SFT}$ | Qwen-3B$^{\text{P\&I}}$ | 10.70% | 33.56% | 21.29% | 52.91% | 15.11% | 0.00% | 59.67% | 32.35% | 14.05% |
| # 19. | Qwen-3B$^{\text{P}}$ $^{GRPO}$ | - | 9.50% | 29.03% | 13.33% | 26.67% | 12.33% | 0.00% | 58.00% | 26.47% | 11.05% |
| # 20. | Qwen-3B$^{\text{P}}$ $^{GRPO}$ | Qwen-3B$^{\text{P}}$ | 9.85% | 32.00% | 12.78% | 28.00% | 12.33% | 0.00% | 61.00% | 29.41% | 12.98% |
| # 21. | Qwen-3B$^{\text{I}}$ $^{GRPO}$ | - | 0.17% | 9.67% | 36.67% | 48.89% | 8.20% | 0.00% | 50.00% | 20.59% | 7.75% |
| # 22. | Qwen-3B$^{\text{I}}$ $^{GRPO}$ | Qwen-3B$^{\text{I}}$ | 0.66% | 13.48% | 37.60% | 51.00% | 8.07% | 0.00% | 52.33% | 23.53% | 9.88% |
| # 23. | Qwen-3B$^{\text{P\&I}}$ $^{GRPO}$ | - | 14.00% | 34.50% | 36.67% | 60.00% | 19.18% | 0.00% | 55.00% | 29.41% | 12.10% |
| # 24. | Qwen-3B$^{\text{P\&I}}$ $^{GRPO}$ | Qwen-3B$^{\text{P\&I}}$ | 14.11% | 39.49% | 35.66% | 61.99% | 19.45% | 0.00% | 57.33% | 35.29% | 15.80% |
| **Qwen2.5-VL-7B-Instruct Fine-tuned Series** | | | | | | | | | | | |
| # 25. | Qwen-7B$^{\text{P}}$ $^{SFT}$ | - | 3.67% | 36.13% | 13.33% | 19.33% | 15.07% | 2.00% | 66.00% | 38.24% | 15.35% |
| # 26. | Qwen-7B$^{\text{P}}$ $^{SFT}$ | Qwen-7B$^{\text{P}}$ | 3.40% | 38.67% | 13.57% | 22.15% | 15.31% | 2.00% | 67.33% | 41.18% | 17.50% |
| # 27. | Qwen-7B$^{\text{I}}$ $^{SFT}$ | - | 0.67% | 9.33% | 12.22% | 50.00% | 13.70% | 1.00% | 64.00% | 23.53% | 8.21% |
| # 28. | Qwen-7B$^{\text{I}}$ $^{SFT}$ | Qwen-7B$^{\text{I}}$ | 0.82% | 13.45% | 11.34% | 51.50% | 13.49% | 1.00% | 66.67% | 26.47% | 10.36% |
| # 29. | Qwen-7B$^{\text{P\&I}}$ $^{SFT}$ | - | 7.50% | 42.67% | 20.00% | 63.33% | 15.07% | 2.00% | 67.00% | 41.18% | 16.55% |
| # 30. | Qwen-7B$^{\text{P\&I}}$ $^{SFT}$ | Qwen-7B$^{\text{P\&I}}$ | 7.95% | 44.85% 🏅 | 21.41% | 65.38% | 14.88% | 2.00% | 69.00% 🏅 | 44.12% | 19.33% |
| # 31. | Qwen-7B$^{\text{P}}$ $^{GRPO}$ | - | 13.00% | 40.00% | 16.67% | 28.89% | 13.70% | 0.00% | 66.00% | 41.18% | 16.10% |
| # 32. | Qwen-7B$^{\text{P}}$ $^{GRPO}$ | Qwen-7B$^{\text{P}}$ | 13.18% | 43.66% | 15.68% | 30.00% | 13.91% | 0.00% | 69.00% 🏅 | 44.12% | 18.88% |
| # 33. | Qwen-7B$^{\text{I}}$ $^{GRPO}$ | - | 0.67% | 8.33% | 26.67% | 86.67% | 16.44% | 0.00% | 60.00% | 23.53% | 7.95% |
| # 34. | Qwen-7B$^{\text{I}}$ $^{GRPO}$ | Qwen-7B$^{\text{I}}$ | 0.67% | 13.04% | 27.82% | 89.31% | 16.44% | 0.00% | 63.00% | 26.47% | 10.15% |
| # 35. | Qwen-7B$^{\text{P\&I}}$ $^{GRPO}$ | - | 14.63% | 43.17% | 13.33% | 90.00% 🏅 | 19.18% | 0.00% | 69.00% 🏅 | 45.61% 🏅 | 17.05% |
| # 36. | Qwen-7B$^{\text{P\&I}}$ $^{GRPO}$ | Qwen-7B$^{\text{P\&I}}$ | 14.92% | 45.56% 🏅 | 13.12% | 93.33% 🏅 | 18.96% | 0.00% | 71.00% 🏅 | 47.06% 🏅 | 21.20% |

The main results is shown in Table 1[5], provide a comprehensive evaluation of the **ICPRL** framework and its components. The analysis validates our core hypotheses regarding in-context physical

---

[4] Although similarly named, the environments pose distinct challenges: *PHYRE* centers on single-step actions that trigger complex chain reactions, while *I-PHYRE* requires multi-step planning with precise temporal and sequential control.

[5] Unlike the results in the original DeepPHY paper, we report S.R. on the test set for *PHYRE* and *I-PHYRE* environments. The results for all other environments are identical to those in DeepPHY.

reinforcement learning, the role of world models, and the importance of training paradigms for generalization in complex physical reasoning tasks.

Our finetuned methods, including both SFT and GRPO, significantly improve the performance of the utilized base open-source VLM, Qwen2.5-VL. The premier configuration, **Qwen-7B$^{\text{P\&I GRPO}}$** paired with **Qwen-7B$^{\text{P\&I}}$** (Row #36), consistently achieves state-of-the-art or comparable performance across nearly all tasks. This result validates our central hypothesis: the synergy between an adaptive policy ($\pi_\theta$), trained via GRPO on multi-episode interaction histories, and an independently-trained world model ($\mathcal{M}_\phi$), that guides planning, is critical for mastering complex physical challenges. Notably, this model not only excels in the trained environments (*PHYRE* and *I-PHYRE*) but also exhibits remarkable generalization to entirely unseen tasks in the other four diverse settings, demonstrating the acquisition of robust and transferable physical intuition.

A direct comparison between models trained with SFT (*e.g.,* Row #29) and those trained with GRPO (*e.g.,* Row #35) reveals the clear superiority of the latter. While SFT enables learning from expert trajectories, GRPO's online policy optimization teaches the agent to dynamically **adapt its strategy in-context** by conditioning on a history of successes and failures. This capability is fundamental to the ICRL paradigm, resulting in a more robust and adaptive policy that is particularly effective in tasks that inherently involve trial-and-error and iterative problem-solving.

The contribution of World Model ($\mathcal{M}_\phi$) is pivotal, transforming the agent from a purely reactive decision-maker into a deliberative planner. Across all finetuned pairs in the table, the full ICPRL configuration (Policy + World Model) consistently outperforms the policy-only variant. This confirms that the world model, acting as an in-context physical simulator, provides crucial foresight. The resulting "propose-verify-select" mechanism, where the policy generates candidate actions and the world model guides a PUCT search to select the most promising one, elevates the agent's reasoning from simple reaction to improved look-ahead planning. Conversely, employing a generic, non-finetuned VLM as a world model (comparing pairs from Rows #5 vs. #6 to #11 vs. #12) degrades performance. This finding, consistent with the original DeepPHY, underscores that general-purpose VLMs currently lack fine-grained interactive physical reasoning capabilities.

Moreover, models trained jointly on both *PHYRE* and *I-PHYRE* (marked as **P&I**) demonstrate superior performance on unseen tasks, when compared to models trained on either environment in isolation (*cf.* Rows #31, #33, and #35). Specifically, the **Qwen-7B$^{\text{P\&I GRPO}}$** policy (Row #35) outperforms its single-environment counterparts across three unseen testbeds. This supports our hypothesis that exposure to diverse physical dynamics enables the model to internalize a more generalizable "policy-improvement operator," facilitating the transfer of learned physical intuition to new domains.

In the *Kinetix* environment, however, our models do not exhibit the same magnitude of performance gain as seen in the other environments. We attribute this to its unique nature, which demands precise fine-grained control of sub-object components and their interactions. In contrast, the other environments primarily involve reasoning about the holistic behavior and trajectory of whole objects.

### 4.3 Ablation Studies

Our primary SFT models are trained on a dataset of successful solution trajectories from expert models (Gemini-2.5-Pro, GPT-4o, etc., detailed in Appendix B), which often include multiple attempts (typically 5–10) before reaching a solution. This strategy is founded on the hypothesis that exposure to this trial-and-error process, even within a supervised framework, enables the model to internalize an iterative problem-solving strategy, aligning with the principles of ICRL.

To validate this hypothesis, we conduct a controlled ablation study detailed in Table 2. We compare our standard SFT models against ablated variants (denoted with a 'single' subscript). These ablated models are trained exclusively on a dataset of first-attempt successful trajectories generated via a systematic enumeration process. To ensure the comparison is fair, the number of training samples was kept identical for both SFT methods within the same environment.

While the 'single' models demonstrate strong initial performance (Att. 1), they often plateau, a behavior particularly pronounced on the *I-PHYRE* benchmark. In contrast, models trained on multi-attempt data exhibit a significantly steeper improvement curve, ultimately achieving superior performance by the final attempts (Att. 10). This confirms that learning from a history of failures and

Table 2: **Ablation Study:** Performance of Multi-Attempt vs. Single-Attempt Training Trajectories.

(a) *PHYRE* Benchmark.

| Policy Model Only | Avg. Att. | Att. 1 | Att. 4 | Att. 7 | Att. 10 |
|---|---|---|---|---|---|
| *MOCK* | 5.00 | 1.50% | 5.87% | 8.60% | 10.80% |
| Qwen-3B | 3.98 | 0.17% | 4.69% | 5.67% | 5.85% |
| Qwen-3B$^{P_{single}\ SFT}$ | 2.60 | 7.63% | 17.20% | 19.83% | 20.97% |
| Qwen-3B$^{P\ SFT}$ | 2.89 | 1.33% | 16.58% | 20.33% | 25.58% |
| Qwen-3B$^{P\ GRPO}$ | 3.23 | 9.50% | 25.13% | 28.80% | 29.03% |
| Qwen-7B | 5.10 | 0.67% | 5.33% | 8.17% | 10.10% |
| Qwen-7B$^{P_{single}\ SFT}$ | 2.61 | 13.17% | 21.58% | 26.97% | 29.14% |
| Qwen-7B$^{P\ SFT}$ | 3.74 | 3.67% | 27.33% | 34.20% | 36.13% |
| Qwen-7B$^{P\ GRPO}$ | 5.70 | 13.00% | 33.17% | 41.77% | 40.00% |
| Qwen-32B | 3.97 | 0.03% | 3.10% | 6.93% | 8.70% |
| Qwen-72B | 4.48 | 2.43% | 9.53% | 12.40% | 14.92% |

(b) *I-PHYRE* Benchmark.

| Policy Model Only | Avg. Att. | Att. 1 | Att. 4 | Att. 7 | Att. 10 |
|---|---|---|---|---|---|
| *MOCK* | 3.81 | 23.33% | 43.33% | 46.67% | 53.33% |
| Qwen-3B | 2.87 | 13.33% | 16.67% | 16.67% | 16.67% |
| Qwen-3B$^{I_{single}\ SFT}$ | 2.97 | 33.43% | 33.43% | 34.54% | 34.54% |
| Qwen-3B$^{I\ SFT}$ | 3.72 | 23.33% | 30.00% | 43.33% | 45.56% |
| Qwen-3B$^{I\ GRPO}$ | 3.81 | 36.67% | 43.33% | 45.56% | 48.89% |
| Qwen-7B | 2.20 | 32.42% | 32.42% | 32.42% | 32.42% |
| Qwen-7B$^{I_{single}\ SFT}$ | 3.44 | 35.56% | 35.56% | 35.56% | 35.56% |
| Qwen-7B$^{I\ SFT}$ | 3.53 | 12.22% | 46.67% | 50.00% | 50.00% |
| Qwen-7B$^{I\ GRPO}$ | 4.37 | 26.67% | 43.33% | 53.33% | 86.67% |
| Qwen-32B | 1.40 | 0.17% | 5.60% | 5.60% | 5.60% |
| Qwen-72B | 2.00 | 13.33% | 30.00% | 43.33% | 43.33% |

recoveries is crucial for developing a more robust policy. Furthermore, the success rate for our models trained on multi-attempt trajectories consistently and monotonically increases as the interaction history lengthens. A compelling trend is found where the average number of attempts for successful solutions (Avg. Att.) tends to increase in tandem with the model's capability (*e.g.,* from **SFT** to **GRPO** variants). This suggests that more advanced models are not simply more efficient, but are capable of tackling more complex problems that inherently require a longer iterative process. This provides direct empirical evidence that the models have learned to leverage ICL to interact with the environment and parse physical intuition. Finally, the outstanding performance of the **GRPO** models further underscores the efficacy of our approach.

Table 3: **Ablation Study:** Impact of Dynamic Visual Feedback on World Model Training.

(a) *PHYRE* Benchmark.

| Policy Model | World Model | Att. 1 | Att. 10 |
|---|---|---|---|
| Qwen-3B$^{P\ SFT}$ | - | 1.33% | 25.58% |
| Qwen-3B$^{P\ SFT}$ | Qwen-3B$^{P}_{w/o\ 5\ Frames}$ | 1.40% | 27.19% |
| Qwen-3B$^{P\ SFT}$ | Qwen-3B$^{P}$ | 1.52% | 28.69% |
| Qwen-3B$^{P\ GRPO}$ | - | 9.50% | 29.03% |
| Qwen-3B$^{P\ GRPO}$ | Qwen-3B$^{P}_{w/o\ 5\ Frames}$ | 9.95% | 30.20% |
| Qwen-3B$^{P\ GRPO}$ | Qwen-3B$^{P}$ | 9.85% | 32.00% |
| Qwen-7B$^{P\ SFT}$ | - | 3.67% | 36.13% |
| Qwen-7B$^{P\ SFT}$ | Qwen-7B$^{P}_{w/o\ 5\ Frames}$ | 3.40% | 37.47% |
| Qwen-7B$^{P\ SFT}$ | Qwen-7B$^{P}$ | 3.40% | 38.67% |
| Qwen-7B$^{P\ GRPO}$ | - | 13.00% | 40.00% |
| Qwen-7B$^{P\ GRPO}$ | Qwen-7B$^{P}_{w/o\ 5\ Frames}$ | 12.68% | 41.76% |
| Qwen-7B$^{P\ GRPO}$ | Qwen-7B$^{P}$ | 13.18% | 43.66% |

(b) *I-PHYRE* Benchmark.

| Policy Model | World Model | Att. 1 | Att. 10 |
|---|---|---|---|
| Qwen-3B$^{I\ SFT}$ | - | 23.33% | 45.56% |
| Qwen-3B$^{I\ SFT}$ | Qwen-3B$^{I}_{w/o\ 5\ Frames}$ | 22.81% | 45.04% |
| Qwen-3B$^{I\ SFT}$ | Qwen-3B$^{I}$ | 23.21% | 47.34% |
| Qwen-3B$^{I\ GRPO}$ | - | 36.67% | 48.89% |
| Qwen-3B$^{I\ GRPO}$ | Qwen-3B$^{I}_{w/o\ 5\ Frames}$ | 36.97% | 47.29% |
| Qwen-3B$^{I\ GRPO}$ | Qwen-3B$^{I}$ | 37.60% | 51.00% |
| Qwen-7B$^{I\ SFT}$ | - | 12.22% | 50.00% |
| Qwen-7B$^{I\ SFT}$ | Qwen-7B$^{I}_{w/o\ 5\ Frames}$ | 11.34% | 50.30% |
| Qwen-7B$^{I\ SFT}$ | Qwen-7B$^{I}$ | 11.34% | 51.50% |
| Qwen-7B$^{I\ GRPO}$ | - | 26.67% | 86.67% |
| Qwen-7B$^{I\ GRPO}$ | Qwen-7B$^{I}_{w/o\ 5\ Frames}$ | 27.12% | 87.51% |
| Qwen-7B$^{I\ GRPO}$ | Qwen-7B$^{I}$ | 27.82% | 89.31% |

Table 3 evaluates the importance of providing the World Model with visual information about the dynamic consequences of an action during its training phase. We compare the full model against a variant where the World Model was trained without the five uniformly sampled post-action frames (detailed in Appendix D). Across both *PHYRE* and *I-PHYRE*, and for all policy model configurations, the full ICPRL framework—which leverages a world model trained with post-action visual frames—consistently outperforms the variant employing an ablated world model (data curated by `w/o 5 Frames`). This performance delta provides strong evidence that incorporating explicit visual feedback of an action's dynamic consequences is crucial for training an effective world model. Which, in turn, enhances its predictive fidelity, directly translating to more effective guidance for the PUCT search procedure at inference time. Furthermore, it is noteworthy that even the ablated world model generally provides some performance lift over the policy-only baseline, underscoring the fundamental utility of our decoupled two-component architecture for deliberative planning.

## 5 CONCLUSION

In this work, we introduced **ICPRL** (In-Context Physical Reinforcement Learning), a framework designed to address the limitations of existing Vision-Language Models in interactive reasoning within dynamic physical environments. Our framework integrates an adaptive policy model with a world model that provides explicit physical intuition. Our extensive evaluations on the diverse DeepPHY

benchmark demonstrate that ICPRL not only achieves significant performance gains over strong baselines but, crucially, maintains robust generalization, enables genuine in-context acquisition of an environment's physical dynamics directly from interactive experience.

While ICPRL represents a significant step forward, this work also highlights promising avenues for future research. First, the policy and world models in the current framework are trained independently. Future work could explore synergistic training paradigms where the models are co-trained, allowing data generated by one to inform and enhance the learning process of the other, potentially creating a virtuous cycle of improvement. Second, our analysis revealed that while ICPRL excels at learning overarching physical principles, its performance gains were less pronounced on tasks like in the *Kinetix* environment, which demands high-dexterity, component-level manipulation. This distinction suggests that such fine-grained control problems may constitute a distinct class of challenges, representing another promising direction for future investigation.

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

## A  BENCHMARK TASK SPECIFICATIONS

In this section, we provide the formal definitions of our evaluation metrics and a comprehensive specification of the task setups across all six environments in the DeepPHY benchmark Xu et al. (2025). Although the task details are documented in the original DeepPHY paper, we restate them here to facilitate a clearer and more accessible understanding of our specific task setups, particularly concerning decision horizons, action granularity, and success criteria.

### A.1  FORMAL DEFINITIONS

To clarify the structure of our interactive evaluation, we define the following terms:

- **Episode (Task Instance):** An episode refers to the complete problem-solving process for a single specific level or puzzle configuration (e.g., PHYRE template 00002:001). An episode concludes when the agent successfully solves the task or exhausts the maximum allowed attempts.

- **Attempt (Trial):** An episode consists of a sequence of up to $K$ attempts.
  - At the start of attempt $k$, the agent receives the visual observation and the text history of previous failed trajectories $H_{k-1} = \{\tau_1, ..., \tau_{k-1}\}$ to perform in-context learning.
  - For PHYRE and I-PHYRE (training environments), the limit is $K = 10$. For evaluation environments like Pooltool and Angry Birds, limits are set according to Table 5.

- **Action Horizon (Per Attempt):** This defines the temporal complexity of a single plan within one attempt.
  - *Single-step / In-advance:* The agent outputs a complete static plan (e.g., placing one ball in PHYRE) or a sequence of timed actions (I-PHYRE) at the start. The simulator then executes the physics until stability.
  - *Sequential / On-the-fly:* The agent interacts with the environment in a turn-based manner (e.g., Kinetix, Angry Birds), observing intermediate states between actions.

### A.2  ENVIRONMENT SPECIFICATIONS

We provide a detailed summary of the input modalities, action spaces, and success criteria for all environments in Table 4 & 5.

Table 4: Detailed Specifications of DeepPHY Environments (Part I): Input Modalities and Action Spaces.

| Environment | Input Modality | Action Space Format |
|---|---|---|
| **PHYRE** | Image + 8×8 Grid Overlay | Discretized Selection: `Cell: [1-64], Radius: [1-8]` |
| **I-PHYRE** | Image + Index IDs | JSON Sequence: `[{"time": t, "index": i}, ...]` |
| **Kinetix** | Image + Motor/Thruster IDs | JSON Vector: `[m_1, m_2, ..., t_1, t_2]` |
| 🎱 **Pooltool** | 2D Top-down View | Discretized Selection: `Speed: [Low/Med/High], Strikespot: [Spin]` |
| 🐦 **Angry Birds** | Screenshot (Annotated) | Code: `[shoot(angle=int, power=float)]` |
| 🐸 **Cut the Rope** | Screenshot (Annotated) | Code: `[cut_pin(id)], [pop_bubble(id)]`, etc. |

Table 5: Detailed Specifications of DeepPHY Environments (Part II): Horizons and Success Criteria.

| Environment | Horizon | Success Criteria ($r_{GT} = 1$) |
|---|---|---|
| **PHYRE** | Single-step | Green ball touches Target. |
| **I-PHYRE** | Sequence | All Red balls fall into abyss. |
| **Kinetix** | Multi-step (Max 16) | Green shape touches Blue, avoids Red. |
| 🎱 **Pooltool** | Multi-step (Max 15) | Pot the 9-ball legally. |
| 🐦 **Angry Birds** | Multi-step (Birds avail.) | All pigs are eliminated. |
| 🐸 **Cut the Rope** | Multi-step | Candy reaches Om Nom's mouth. |

## A.3 CONCRETE IMPLEMENTATION EXAMPLE: CUT THE ROPE

To illustrate how continuous gaming environments are adapted for VLM control, we detail the implementation of the *Cut the Rope* task.

**Visual Input:** The model receives a screenshot where interactive elements are annotated with numerical IDs. For example, a rope anchored to a pin might be labeled "Pin 1", and a floating bubble containing candy might be labeled "Bubble 2".

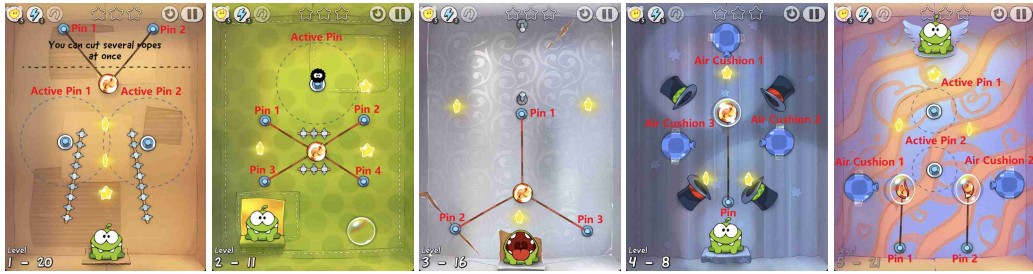

Figure 3: Examples of static element annotation in the *Cut the Rope* game. The figure displays screenshots from various levels where key static props—such as Pins, Active Pins, and Air Cushions—are clearly marked with numerical IDs. This method converts pixel-level visual information, enabling Agent to accurately identify and locate each key object within the game world.

**Action Space:** Instead of continuous gestures, the model outputs structured Python-like function calls within square brackets. The interpreter parses these tokens into game actions. Supported commands include:

- `[cut_pin(id=3)]`: Cuts the rope attached to Pin #3.
- `[pop_bubble(id=2)]`: Pops the bubble labeled #2 to drop the candy.
- `[tap_air_cushion(id=1, times=2)]`: Taps the air cushion #1 twice to blow air.
- `[sleep(seconds=0.5)]`: Waits for 0.5s. This is critical for handling swing dynamics (e.g., cutting a rope only when the candy swings to a specific angle).

## B MODEL ABBREVIATIONS

The abbreviations used throughout the paper and the full details of the corresponding open-source and closed-source models are listed in Table 6.

Table 6: List of Models Abbreviations

| Model | | Abbreviation |
|---|---|---|
| *MOCK* | | *MOCK* |
| **Open-Source Models** | | |
| **Qwen2.5-VL Series** | | |
| Qwen2.5-VL-3B-Instruct | | Qwen-3B |
| Qwen2.5-VL-7B-Instruct | | Qwen-7B |
| Qwen2.5-VL-32B-Instruct | | Qwen-32B |
| Qwen2.5-VL-72B-Instruct | | Qwen-72B |
| **Closed-Source Models** | | |
| Claude 4.0 Opus | (Anthropic, 2025) | Claude 4.0 Opus |
| Gemini-2.5-Pro-06-17 | (Comanici & et al., 2025) | Gemini-2.5-Pro |
| GPT-o3-0416 | (OpenAI, 2025) | GPT-o3 |

To clearly distinguish the fine-tuned models developed in our work, we adopt a systematic naming convention, which we illustrate using the Qwen-3B model as example. The names for the Qwen-7B-

based models follow the same convention. We use GREEN to denote Policy Models and ORANGE for World Models.

**Policy Models.** The nomenclature for our Policy Models follows the format: $\text{BaseModel}^{\text{Env}_{\text{Data-Type}}\ Method}$. Each component in the naming structure is defined as follows:

- **Env:** This component begins by indicating the environment(s) used in fine-tuning:
    - **P**: Denotes fine-tuning exclusively on *PHYRE*.
    - **I**: Denotes fine-tuning exclusively on *I-PHYRE*.
    - **P&I**: Denotes a model trained using tasks from both the *PHYRE* and *I-PHYRE* environments.
- **Method:** This term specifies the training algorithm used:
    - **SFT**: Supervised Fine-Tuning.
    - **GRPO**: Group Relative Policy Optimization (Shao et al., 2024).
- **Data-Type:** An optional subscript for the environment term specifies the nature of the training data:
    - **(No subscript)**: The model was trained on a dataset of successful solution trajectories collected from expert models - Gemini-2.5-Pro (Comanici & et al., 2025), GPT-4o, and GPT-4o-mini (OpenAI, 2024). These trajectories were often generated over multiple attempts (typically 5–10) to find a solution. This strategy is designed to enable the model to learn from a process of trial and error, even within a supervised framework. Our experimental results support this methodology, showing it is consistent with the philosophy of ICRL.
    - **single**: The model was trained on a dataset of first-attempt successful trajectories. These trajectories were generated via a systematic enumeration process over the action space.

For example, the model named **Qwen-3B**$^{\text{P}_{\text{single}}\ SFT}$ is a policy model based on Qwen2.5-VL-3B-Instruct, trained via SFT, only on single-attempt successful trajectories from the *PHYRE* environment.

**World Models.** The naming for our World Models is more concise. As all world models are SFT. A given name, such as **Qwen-3B**$^{\text{P}}$, directly specifies the base model (Qwen2.5-VL-3B-Instruct) and the environment (*PHYRE*) used for its training.

World models can take one possible subscript: "$_{w/o\ 5\ Frames}$". This subscript, meaning "without 5 frames," represents a crucial detail about its training dataset. As described in Subsection 4.3, this particular world model was trained on a curated dataset that *excluded* the five uniformly sampled post-action video frames.

## C  MATHEMATICAL DEFINITIONS

In this section, we provide the formal mathematical definitions for the components used in the ICPRL framework to ensure reproducibility and clarity.

We formalize the interactive physical reasoning task as a Partially Observable Markov Decision Process (POMDP) augmented with interaction history.

- **State Space ($\mathcal{S}$):** The underlying physical state of the simulator (e.g., positions, velocities, friction coefficients).
- **Observation Space ($\mathcal{O}$):** The visual rendering of the state $o_t \in \mathcal{O}$, which may include annotated overlays (as detailed in Section A & DeepPHY Xu et al. (2025)).
- **Action Space ($\mathcal{A}$):** The set of executable commands. While $\mathcal{A}$ in physics simulators is often continuous, we discretize it into text tokens for VLM processing. For example, in Angry Birds, continuous angles $\theta \in [0, 90]$ are discretized into integer bins.

- **Success Criteria** ($r_{GT}$)**:** A binary reward signal returned by the environment at the end of an episode. $r_{GT} = 1$ denotes success; $r_{GT} = 0$ denotes failure.

- **Trajectory** ($\tau$)**:** A sequence recording a single attempt's interaction: $\tau = (o, a, r_{GT})$.

- **Interaction History** ($H$)**:** A collection of past failure trajectories within the same problem instance: $H_k = (\tau_1, \tau_2, \ldots, \tau_{k-1})$.

- **Policy Mapping:** The policy $\pi_\theta(a|o, H)$ maps the current observation and history to a probability distribution over actions.

- **Reference Policy:** $\pi_{ref}$ is the frozen version of the policy model derived from the Supervised Fine-Tuning (SFT) stage. It serves as the anchor for the KL-divergence penalty in Equation (1), preventing the RL-tuned policy $\pi_\theta$ from deviating excessively from natural language distribution and maintaining the validity of the output format.

## D ICPRL's Algorithms

The World Model $\mathcal{M}_\phi$ Dataset Curation procedure is formalized in Algorithm 1.

---

**Algorithm 1** World Model Dataset Curation.

---

**Require:** Task set $\mathcal{X}$; action space $\mathcal{A}$; simulator SIMULATE$(x, a)$ returning video $\tau$, action end $t_{\text{act}}$, dynamics end $t_{\text{dyn}}$, label $y \in \{0, 1\}$; number of frames $m{=}5$; diversity threshold $\varepsilon$ (optional).
**Ensure:** WM dataset $\mathcal{D}_{\text{WM}}$ of tuples $(x, I_0, I_{\text{ann}}, a, F_{1:m}, y, T)$.
1: $\mathcal{D}_{\text{WM}} \leftarrow \emptyset$
2: **for** each task $x \in \mathcal{X}$ **do**
3:      $I_0 \leftarrow$ initial raw image of $x$; $I_{\text{ann}} \leftarrow$ initial annotated image of $x$
4:      $S \leftarrow$ **all** actions in $\mathcal{A}$ that solve $x$           ▷ Enumerate via simulation or cache
5:      $k \leftarrow |S|$
6:      $F \leftarrow \emptyset$                                  ▷ Collect $k$ diverse, failing actions
7:      **while** $|F| < k$ **do**
8:          Sample $a \sim \mathcal{A} \setminus S$
9:          **if** $a$ is *distinct* from $S \cup F$ (e.g., using distance $\geq \varepsilon$) **then**
10:              $(y', \_, \_, \_) \leftarrow$ SIMULATE$(x, a)$
11:              **if** $y' = 0$ **then**
12:                  $F \leftarrow F \cup \{a\}$
13:              **end if**
14:          **end if**
15:      **end while**
16:      **for** each $a \in S \cup F$ **do**
17:          $(\tau, t_{\text{act}}, t_{\text{dyn}}, y) \leftarrow$ SIMULATE$(x, a)$
18:          Uniformly sample $m$ timestamps in $(t_{\text{act}}, t_{\text{dyn}}]$ and extract frames $F_{1:m}$ from $\tau$
19:          $T \leftarrow$ VLM$(I_0, I_{\text{ann}}, a, F_{1:m}, y)$
20:                  ▷ Prompt VLM for Grounding & World Modeling text, i.e., $p_{\text{pred}}$ ground truth
21:          **if** human verification passes **then**
22:              $\mathcal{D}_{\text{WM}} \leftarrow \mathcal{D}_{\text{WM}} \cup \{(x, I_0, I_{\text{ann}}, a, F_{1:m}, y, T)\}$
23:          **end if**
24:      **end for**
25: **end for**
26: **return** $\mathcal{D}_{\text{WM}}$

---

The inference procedure with Root-Node Search in ICPRL is detailed in Algorithm 2.

## E COMPUTATION OF STABILITY AND UNCERTAINTY FOR PUCT SEARCH

This section provides a detailed specification of the methods used to instantiate the stability score ($p_{\text{stab}}$) and the confidence-based score (score$(a|o)$) for the PUCT search procedure, as outlined in Subsection 3.3 and Algorithm 2. The choice of method is tailored to the distinct characteristics of the action spaces across the various environments within the DeepPHY benchmark (Xu et al., 2025).

918
919
920
921
922
923
924
925
926

---

**Algorithm 2** Inference with Root-Node Search.

---

**Require:** Observation $o$; policy $\pi_\theta$; world model $\mathcal{M}_\phi$; hyperparameters $S, K, B, \lambda, c_{\text{PUCT}}$.
**Ensure:** Best action $a^*$.

1:                                            ▷ **Stage 1: Candidate Generation and Prior**
2:  $A_{\text{samples}} \leftarrow \emptyset$
3: **for** $i = 1 \to S$ **do**
4:     Sample $a \sim \pi_\theta(\cdot \mid o)$ and add to $A_{\text{samples}}$
5: **end for**
6:  $A \leftarrow \text{UNIQUE}(A_{\text{samples}})$
7: **for** each action $a \in A$ **do**
8:     $P(a) \leftarrow \text{COUNT}(a, A_{\text{samples}})/S$
9: **end for**
10:                             ▷ **Stage 2 & 3: PUCT Search Guided by WM Scores**
11: $N(a) \leftarrow 0, Q(a) \leftarrow 0$ for all $a \in A$
12: $last\_a \leftarrow \text{null}, consecutive\_choices \leftarrow 0$
13: **for** $t = 1 \to B$ **do**
14:     $N_{\text{tot}} \leftarrow \sum_{b \in A} N(b)$
15:     $a_t \leftarrow \arg\max_{a \in A} \left[ Q(a) + c_{\text{PUCT}} \cdot P(a) \cdot \frac{\sqrt{N_{\text{tot}}}}{1 + N(a)} \right]$
16:                                 ▷ Early stopping check
17:     **if** $a_t = last\_a$ **then** $consecutive\_choices \leftarrow consecutive\_choices + 1$
18:     **else** $consecutive\_choices \leftarrow 1$
19:     **end if**
20:     **if** $consecutive\_choices \geq 3$ **then break**
21:     **end if**
22:     $last\_a \leftarrow a_t$
23:                            ▷ Get WM score by querying K times
24:     $p_{\text{succ\_list}} \leftarrow [], p_{\text{stab\_list}} \leftarrow []$
25:     **for** $j = 1 \to K$ **do**
26:         $\hat{p}_{\text{succ}}^{(j)} \leftarrow \mathcal{M}_\phi(o, a_t)$
27:         $\hat{p}_{\text{stab}}^{(j)} \leftarrow \mathbb{E}_{a' \sim D(\mathcal{B}_\delta(a_t))}[\mathcal{M}_\phi(o, a').\hat{p}_{\text{succ}}]$         ▷ Estimated via jitters
28:         Append $\hat{p}_{\text{succ}}^{(j)}$ to $p_{\text{succ\_list}}$; $\hat{p}_{\text{stab}}^{(j)}$ to $p_{\text{stab\_list}}$
29:     **end for**
30:     $\mu_p \leftarrow \text{MEAN}(p_{\text{succ\_list}})$; $\mu_s \leftarrow \text{MEAN}(p_{\text{stab\_list}})$
31:     $s_t \leftarrow (1 - \lambda)\mu_p + \lambda\mu_s$
32:                              ▷ Update search statistics
33:     $Q(a_t) \leftarrow (Q(a_t) \cdot N(a_t) + s_t)/(N(a_t) + 1)$
34:     $N(a_t) \leftarrow N(a_t) + 1$
35: **end for**
36:                                    ▷ **Stage 4: Final Selection**
37: $a^* \leftarrow \arg\max_{a \in A} Q(a)$
38: **return** $a^*$

---

We categorize our approach into two main strategies, summarized in Table 7: direct action-space perturbation (for environments with well-defined action neighborhoods), and model confidence estimation (for environments highly sensitive to minimal action changes).

Table 7: Calculation Methods of Stability and Uncertainty for PUCT Search.

| Environment | Methodology | Action Space | Perturbation / Sampling Details |
|---|---|---|---|
| **Strategy 1: Action Space Perturbation for Stability Estimation** | | | |
| *Stability Score:* $\hat{p}_{\text{stab}}(a) = \frac{1}{J}\sum_{j=1}^{J}\mathbb{I}[\mathcal{M}_\phi(o, a'_j) \to success]$ | | | |
| **PHYRE** | Action Space Perturbation | $(x, y, r)$ Grid coord. & radius | Perturb $(x, y)$ by $\pm 1$ (4 directions), and $r$ by $\pm 1$ or 0. |
| **I-PHYRE** | | Sequence of timed events $[(i_k, t_k)]$ | Temporal jittering: perturb timestamps $t_k$ with $\pm\Delta t \in \{\pm 0.5\text{s}, \pm 1.0\text{s}\}$ for a subset of events. |
| **Angry Birds** | | $(\theta, p)$ Angle & power | Perturb $\theta$ with $\Delta_\theta \in \{-5°, 0°, 5°\}$ and $p$ with $\Delta_p \in \{-0.1, 0, 0.1\}$. |
| **Strategy 2: Confidence-Based Scoring for Highly Sensitive Environments** | | | |
| *Uncertainty Score (LCB):* $score(a\|o) = \mu_p - \beta\sigma_p$ | | | |
| **Kinetix** | Model Confidence (LCB) | Varies (e.g., timings, forces, positions) | $K = 8$ stochastic forward passes of $\mathcal{M}_\phi(o, a)$ using different temperatures sampled from $[0.1, 1.0]$. This yields a set of probabilities $\{p^{(j)}\}_{j=1}^{K}$. |
| **Pooltool** | | | |
| **Cut the Rope** | | | |

### E.1 ACTION SPACE PERTURBATION FOR STABILITY ESTIMATION

For environments where the action space allows for meaningful local perturbations, we estimate action stability by sampling a neighborhood of actions around a candidate action $a$. The world model, $\mathcal{M}_\phi$, evaluates the success probability for each perturbed action. The stability is then defined as the fraction of these neighboring actions predicted to be successful.

$$\hat{p}_{\text{stab}}(s, a) = \frac{1}{J}\sum_{j=1}^{J}\mathbb{I}[\mathcal{M}_\phi(o, a'_j) \to \text{success}],$$

where $\{a'_j\}_{j=1}^{J}$ is the set of $J$ perturbed actions in the neighborhood of $a$. An action is deemed robustly successful if its stability score meets or exceeds a threshold of 0.75.

**PHYRE** (Bakhtin et al., 2019). The action space is defined by a grid cell selection $(x, y)$ and a radius level $r$. For a given action, we generate a neighborhood of $J = 12$ perturbations by perturbing its grid coordinates by one unit in each of the four cardinal directions and simultaneously varying its radius level by $\pm 1$ or keeping it unchanged. Any resulting perturbations that fall outside the defined grid boundaries or violate the valid radius range are discarded.

**I-PHYRE** Li et al. (2024). For this environment, an action $a$ is a sequence of timed events, $a = [(i_1, t_1), (i_2, t_2), \ldots, (i_L, t_L)]$, where $i_k$ is the index of a block to be eliminated at absolute time $t_k \in [0, T_{\max}]$, with $t_k$ being non-decreasing. Perturbations are introduced via temporal "jittering". For a given event $k$, its timestamp is perturbed as follows:

$$t'_k = \text{clip}(t_k + \delta, 0, T_{\max}),$$

where $\delta \in \{-\Delta t, +\Delta t\}$. In our experiments, we use $\Delta t \in \{0.5\text{s}, 1.0\text{s}\}$. To preserve the intended causal sequence, if a jittered timestamp $t'_k$ violates the non-decreasing order (e.g., $t'_k < t_{k-1}$), it is projected back into the valid interval $[t_{k-1} + \epsilon, t_{k+1} - \epsilon]$ for a small positive constant $\epsilon$ (e.g., $10^{-3}$s). For each candidate action, we generate a neighborhood of perturbed sequences by randomly selecting a subset (#2 in our experiments) of its timed events and applying these jitters.

**Angry Birds**[6]. The action $a = (\theta, p)$ is defined by a launch angle $\theta \in [0°, 90°]$ and a power level $p \in [0, 1]$. We construct a neighborhood of $J = 9$ actions by applying discrete perturbations to both parameters: $a' = (\theta + \Delta_\theta, p + \Delta_p)$, where $\Delta_\theta \in \{-5°, 0°, 5°\}$ and $\Delta_p \in \{-0.1, 0, 0.1\}$. All perturbed values are clipped to their valid ranges:

$$\theta' \leftarrow \text{clip}(\theta', 0°, 90°),$$
$$p' \leftarrow \text{clip}(p', 0, 1).$$

---

[6]https://apps.apple.com/us/app/rovio-classics-angry-birds/id1596736236

Duplicate actions resulting from clipping at the boundaries are removed.

### E.2 CONFIDENCE-BASED SCORING FOR HIGHLY SENSITIVE ENVIRONMENTS

In environments such as ***Kinetix*** (Matthews et al., 2025), 🎱 ***Pooltool*** (Kiefl, 2024), and 🐸 ***Cut the Rope***[7], the dynamics can be chaotic and the success of an action often hinges on precise execution. In these cases, even minimal perturbations can lead to drastically different outcomes, rendering action-space stability a less informative metric.

Instead, we assess the world model's **confidence** in its own prediction for a given action. For a single observation-action pair $(o, a)$, we perform $K$ stochastic forward passes through world model $\mathcal{M}_\phi$, by sampling from a range of temperatures (in our experiments, $K = 8$ samples with temperatures drawn uniformly from $[0.1, 1.0]$). This yields a set of success probability predictions:

$$\{p^{(1)}, p^{(2)}, \ldots, p^{(K)}\}, \quad p^{(j)} \in [0, 1].$$

We aggregate these predictions into a single score using a Lower Confidence Bound (LCB) formulation, which is then supplied as the $Q(a)$ value to the PUCT algorithm. The score is computed as:

$$\text{score}(a|o) = \mu_p - \beta \cdot \sigma_p,$$

where $\mu_p = \frac{1}{K} \sum_{j=1}^{K} p^{(j)}$, and $\sigma_p = \sqrt{\frac{1}{K-1} \sum_{j=1}^{K} (p^{(j)} - \mu_p)^2}$. This approach favors actions for which the world model is not only optimistic (high mean $\mu_p$) but also confident (low standard deviation $\sigma_p$). The hyperparameter $\beta > 0$ (set to 0.2 in our experiments) controls the penalty for uncertainty.

## F   LLM USAGE

Our use of LLMs in this paper mainly comprises two parts: writing assistance and tools for data construction. During writing, we use GPT-5 and Gemini-2.5-Pro to help polish the exposition. In the data construction process, as described in Subsection 3.3 and Appendix D, we employ Gemini-2.5-Pro, GPT-4o, and GPT-4o-mini to assist with data generation.

---

[7]https://apps.apple.com/cn/app/cut-the-rope/id1024507512

