# OpenReview forum: "ICPRL: Acquiring Physical Intuition from Interactive Control"
_ICLR.cc/2026/Conference — Submitted to ICLR 2026_

### Official Review · Reviewer_zPwA · 2025-10-30

**Soundness:** 2
**Presentation:** 3
**Contribution:** 2
**Rating:** 4
**Confidence:** 3

**Summary:**

The paper introduces In-Context Physical Reinforcement Learning (ICPRL), a framework that enables vision-language models (VLMs) to perform interactive physical reasoning in dynamic environments without weight updates. The approach combines an adaptive policy model—trained with Generalized Reinforcement Preference Optimization (GRPO) on multi-episode interaction histories—to learn in-context adaptation, with a separately trained world model that predicts action outcomes and guides planning via a PUCT search. Both models are built on Qwen2.5-VL-3B/7B backbones and trained on PHYRE and I-PHYRE, then evaluated on unseen environments (Kinetix, Pooltool, Angry Birds, Cut the Rope) within the DeepPHY benchmark. Experiments show that ICPRL improves the performance significantly compared to the baselines.

**Strengths:**

1. The paper presents an in-context reinforcement learning framework applied to vision-language models (VLMs), leveraging VLMs' inherent generalization and in-context reasoning abilities—an idea that could have notable value for the broader learning and embodied AI community.

2. The incorporation of a world model for score prediction and planning is a useful design that enhances reasoning and supports generalization to unseen domains, offering a potentially transferable mechanism for future adaptive agents.

3. The experimental evaluation is comprehensive, covering multiple physics-based environments across the DeepPHY benchmark. The writing and presentation are clear.

**Weaknesses:**

1. The paper lacks clarity regarding the decision horizon and episode length, making it difficult for readers to assess how well the proposed method generalizes to tasks with longer temporal dependencies.

2. The chosen benchmarks are highly structured and have short decision horizons, which diminishes the reinforcement learning aspect. The policy behavior may resemble an enumerative strategy—selecting unseen actions based on past outcomes—rather than genuine long-horizon reasoning. It remains unclear whether the approach would hold in settings like Atari or DMC that require extended sequential control.

3. It is unclear whether ICPRL genuinely enhances in-context learning capability beyond what is already achieved through supervised fine-tuning (SFT) or task-specific adaptation. In tasks where models have been heavily fine-tuned, it remains questionable whether adding the ICPRL stage provides additional adaptive benefit rather than simply reinforcing prior task-specific behavior.

4. The definition and role of "attempts" are insufficiently detailed, and since this metric is tightly coupled to the benchmark structure, it limits interpretability and reproducibility.

**Questions:**

1. Could the authors clarify the episode length, number of actions per episode, and the formal definition of “attempts” used across different environments?

2. How do the authors rule out the possibility that the improvements arise primarily from adaptation to task format rather than genuine in-context learning (weakness 3)? Including an additional baseline or diagnostic analysis addressing this would strengthen the paper’s claims.

---

> ### Author Response · Authors · 2025-11-21
> **Response to Reviewer zPwA**
>
> We thank the reviewer for their thoughtful assessment and for recognizing the value of our VLM+ICRL framework and the comprehensive evaluation. We appreciate the opportunity to clarify why ICPRL provides a distinct adaptive advantage over SFT.
>
> ### **1. Clarifications on Definitions: Attempts, Episodes, and Horizons**
>
> We adhere strictly to the problem formalization in the **DeepPHY [1]** benchmark. We will include these formal definitions in the Appendix of the revision.
>
> - **Episode (Meta-Episode):** Defined as the complete problem-solving process for a single task instance. An episode consists of a sequence of up to $K$ **Attempts** (Trials) to solve the puzzle.
>     - For **PHYRE** and **I-PHYRE** (our training environments), the limit is **$K=10$**.
>     - At the start of attempt $k$, the agent receives the history of previous failed trajectories $H^{(k)} = \{\tau^{(1)}, \dots, \tau^{(k-1)}\}$ to perform in-context learning.
> - **Action Horizon (Per Attempt):** This defines the complexity of a single plan.
>     - **PHYRE (In-advance Planning):** **Horizon = 1**. The agent outputs a single static action (placing one object), and the simulator executes the subsequent chain reaction.
>     - **I-PHYRE (In-advance Planning):** **Horizon = Variable (Sequence)**. Although the agent outputs the plan in a single turn, the action space is a **JSON sequence of timed interventions** (e.g., `[{time: 0.5, index: 2}, {time: 2.1, index: 0}]`). The agent must reason about the temporal order and causal dependencies of multiple events, not just a single step.
>
> ### **2. Response to "ICPRL vs. SFT" (Key Weakness 3)**
>
> The reviewer raises a critical question: *Does ICPRL genuinely enhance in-context learning, or just reinforce SFT/task format?*
>
> **Our ablation study in Table 2 provides the definitive answer to this.**
> We compared a model trained via **SFT** (on successful trajectories) against our **ICPRL (GRPO)** model.
>
> - **SFT Behavior (Plateau):** As shown in Table 2(a), the `Qwen-7BP SFT` model starts with 3.67% success at Attempt 1 and reaches 36.13% by Attempt 10. Crucially, its improvement curve is flatter. SFT teaches the model *what a correct solution looks like*, but not necessarily *how to fix a wrong one*.
> - **ICPRL Behavior (Adaptation):** The `Qwen-7BP GRPO` model starts higher (13.00%) but, more importantly, **climbs steeper** to 40.00%.
> - **The "Delta" proves the mechanism:** The GRPO objective explicitly optimizes the **expected reward over the history**. It penalizes the model if it repeats the same mistake or fails to leverage the feedback from the previous turn. This forces the model to learn an **error-correction operator** (e.g., "If I missed to the left, I must aim further right"), which SFT does not explicitly enforce.
>
> **Ruling out "Task Format Adaptation":**
> If the improvement were merely due to "adapting to the task format," the model would fail on **unseen tasks** with completely different formats.
>
> - **Evidence:** Our model (trained *only* on PHYRE) generalizes Zero-Shot to **Pooltool** (71.0% success) and **Angry Birds**.
> - These environments have totally different action spaces (cue stick angle vs. slingshot pullback) and visual formats. Mere format adaptation cannot explain this transfer; it indicates the acquisition of generalized **in-context physical intuition**.
>
> ### **3. Response to "Enumerative Strategy" and Benchmarks**
>
> The reviewer suggests the strategy might be "enumerative" and questions the comparison to long-horizon tasks like Atari.
>
> - **Reasoning vs. Control:** We respectfully argue that "Physical Reasoning" presents a different type of complexity than "Sequential Control" (Atari). In DeepPHY, a single action triggers a complex, chaotic chain reaction. The difficulty lies in **predicting dynamics** (Simulation) rather than assigning credit over thousands of timesteps.
> - **Smart Search (Meta-RL):** While the behavior resembles enumeration, it is **"Smart Search"** (Bayesian Optimization in natural language). The agent isn't randomly guessing; it is narrowing down the hypothesis space based on physical feedback. Teaching a VLM to perform this "active search" is a core contribution of ICRL, distinct from the reactive control policies typical in Atari benchmarks.
>
> ### **4. Conclusion**
>
> We believe the combination of **Table 2 (ICRL > SFT)** and **Table 1 (Cross-Domain Generalization)** firmly establishes that ICPRL learns a robust, transferable adaptation strategy that goes beyond simple supervised imitation.
>
> [1] DeepPHY et al. DeepPHY: Benchmarking Agentic VLMs on Physical Reasoning

---

> > ### Comment · Reviewer_zPwA · 2025-11-25
> >
> > I appreciate the authors' efforts in the rebuttal. After revisiting the paper with a clearer understanding of the task setting, many of my earlier concerns have been addressed. I now recognize that the proposed approach offers meaningful contributions, and I will raise my score to 6. I also strongly recommend placing the task description in a more prominent early section. The paper currently presumes familiarity with DeepPHY, and I spent considerable time trying to infer the exact problem setup along with the contribution.
> >
> > That said, the concern regarding long-horizon environments remains unresolved. DeepPHY is not the only domain with complex physical dynamics. Environments such as Habitat [1] and MineRL [2] also involve rich continuous dynamics, and even some Atari games exhibit non-trivial physics. These benchmarks feature significantly longer horizons, and it is unclear whether ICPRL would remain tractable in such settings. As the sequence length grows, cross-episode return optimization will quickly become computationally and algorithmically infeasible. If ICPRL is limited to short-horizon tasks, then its applicability is constrained. Therefore, while I appreciate the clarifications, I cannot further increase my score.
> >
> > ---
> > [1] Szot, Andrew, et al. "Habitat 2.0: Training home assistants to rearrange their habitat." Advances in neural information processing systems 34 (2021): 251-266.
> >
> > [2] Guss, William H., et al. "Minerl: A large-scale dataset of minecraft demonstrations." arXiv preprint arXiv:1907.13440 (2019).

---

### Official Review · Reviewer_KzRK · 2025-10-31

**Soundness:** 3
**Presentation:** 3
**Contribution:** 2
**Rating:** 4
**Confidence:** 4

**Summary:**

This paper introduces ICPRL (In-Context Physical Reinforcement Learning), a framework designed to enable Vision-Language Models (VLMs) to perform interactive reasoning and planning in dynamic physical environments. The core idea is to decouple the agent into two components: an adaptive policy model trained via Group Relative Policy Optimization (GRPO) and a world model trained separately and offline to predict the physical outcomes of actions. At inference, the adaptive policy acts as a prior, proposing candidate actions. The world model then evaluates these candidates, and its predictions are used to guide a PUCT search algorithm to select the optimal action. The authors evaluate this framework on the DeepPHY benchmark.

**Strengths:**

* Valid and Sound Idea: The paper presents a valid and logical approach. Combining the principles of In-Context Reinforcement Learning (ICRL) with an explicit, learned world model is a sensible strategy for tackling complex, interactive physical reasoning tasks that VLMs currently struggle with.
* Effective Paradigm Extension: The framework successfully extends the well-established planning and learning paradigm to the VLM domain. The architecture, which integrates a policy prior, a value/outcome model, and a search procedure, is a proven combination, and the paper shows it can be effectively adapted for VLM-based agents.
* Comprehensive Experiments: The experimental evaluation is a clear strength. The authors test their framework across diverse environments in the DeepPHY benchmark and compare it against strong VLM baselines. The inclusion of thorough ablation studies provides convincing evidence for the contribution of each component of the ICPRL framework.

**Weaknesses:**

* Marginal Novelty over Existing Paradigms: The primary weakness is that the framework's architecture is fundamentally a straightforward extension of the MuZero algorithm, adapted for the VLM setup. MuZero also combines a learned policy prior and a world model to guide a Monte Carlo Tree Search (MCTS, of which PUCT is a variant). While applying this to VLMs is a good engineering contribution, the paper does not offer new scientific insight into the underlying principles of planning or model-based reinforcement learning, as these concepts are already well-proven.
* Limited Impact Given Prior Work: Related to the first point, the Dreamer series of works has also extensively demonstrated that model-based RL (learning a world model and planning within it) is highly effective. This existing body of work makes the core contribution of this paper feel more marginal and incremental.
* Questionable Generalization Claims: The tasks in DeepPHY benchmark were proposed to evaluate generalization. The authors claim strong generalization by training on two environments and testing on the other four. However, it has been shown that standard RL agents can achieve good performance on these tasks, which may dilute the significance of the results. More importantly, the paper lacks sufficient detail on the nature of the generalization being tested.

**Questions:**

1. Scientific Contribution: Given the strong architectural parallels to MuZero, could the authors elaborate on what they see as the key new scientific insight from this work, beyond demonstrating a successful application of a known-good algorithm to the VLM domain?
2. Within-Task Generalization: The paper focuses on cross-task generalization (training on PHYRE/I-PHYRE, testing on others). How does the model perform on within-task generalization? For example, was the model evaluated on new, unseen levels or configurations of the training environments (PHYRE and I-PHYRE) that were held out from the training data?
3. Generalization Evaluation Protocol: Could you clarify the generalization protocol further?

---

> ### Author Response · Authors · 2025-11-21
> **Response to Reviewer KzRK**
>
> We thank the reviewer for the insightful comments.
>
> ### **Response to "Marginal Novelty over MuZero/Dreamer"**
>
> While ICPRL shares the high-level "Model-Based Planning" philosophy with MuZero and Dreamer, there is a fundamental scientific distinction: **MuZero/Dreamer are Tabula Rasa Specialists, whereas ICPRL is a Pre-trained Generalist.**
>
> 1. **Latent vs. Semantic Representation:**
>     - **MuZero/Dreamer:** They learn a *latent* dynamics model from scratch. The learned state representation $s_t$ is mathematically optimized for value prediction in *that specific environment* (e.g., Atari Breakout). This latent state has no semantic meaning and **cannot transfer** to a different environment (e.g., Atari Pong).
>     - **ICPRL:** We leverage the **Semantic and Visual** representations of a VLM. Our World Model does not just predict a reward; it predicts a natural language description of physical evolution. Because "rebound" and "collision" are universal physical concepts grounded in the VLM's pre-training, our model enables **Zero-Shot Transfer** to completely new physics engines (e.g., from PHYRE to Pooltool), which is impossible for standard MuZero without retraining.
> 2. **The Role of "Learning":**
> ICPRL learns *how to reason about physics*. The scientific insight here is that by training on interaction histories (via GRPO), we are not just fitting a value function, but teaching the VLM an **in-context policy improvement algorithm**. This allows the agent to adapt its strategy at inference time without weight updates, a capability absent in the fixed inference policies of MuZero.
>
> ### **Response to "Scientific Contribution"**
>
> Beyond the architectural application, our key scientific insights are:
>
> 1. **Language as a Generalizable State Abstraction for Physics:** We demonstrate that natural language serves as a superior state abstraction for *transferable* physical control than the task-specific latent vectors used in traditional MBRL. This allows physical intuition learned in 2D puzzles (PHYRE) to control complex billiards simulations (Pooltool) zero-shot.
> 2. **Decoupled Acquisition of Intuition and Strategy:** We show that physical intuition (World Model: *What will happen?*) and strategic adaptation (Policy: *How to improve?*) can be learned separately and recombined. The Policy learns a meta-strategy (trial-and-error) that is agnostic to the specific physics, while the World Model provides the domain grounding.
>
> ### **Response to "Generalization Claims & Protocol"**
>
> **Clarification on Protocol:**
> Our protocol is far more rigorous than standard RL benchmarks:
>
> - **Training:** Only on PHYRE and I-PHYRE.
> - **Testing (Within-Task):** On **held-out test sets** of PHYRE/I-PHYRE (unseen levels).
> - **Testing (Cross-Task):** On 4 **completely different games** (Pooltool, Angry Birds, etc.) with distinct visual styles, action spaces, and physics engines.
>
> **Addressing "Standard RL Agents":**
> The reviewer noted that "standard RL agents can achieve good performance." We respectfully clarify that this applies *only* to **Within-Task** settings.
>
> - A standard PPO agent trained on PHYRE achieves 0% success on Angry Birds because its policy is overfitted to PHYRE's observation space and dynamics.
> - In contrast, **ICPRL achieves 71.0% on Pooltool (unseen)**. This **Cross-Domain Generalization** is the core innovation and is *not* achievable by standard RL agents.
>
> ### **Performance on Within-Task Generalization?**
>
> Yes. In **Table 1**, the results for PHYRE and I-PHYRE are reported on the **official test sets** (unseen levels/configurations), following the DeepPHY protocol. Our method achieves **93.3%** on I-PHYRE (Test), significantly outperforming the baselines.
>
> ### **Clarify Generalization Protocol.**
>
> We will add a  dedicated "Evaluation Protocol" subsection in the revision:
>
> 1. **Source Domains:** Train Policy/World Model on PHYRE + I-PHYRE.
> 2. **Target Domains:** Zero-shot inference on Kinetix, Pooltool, Angry Birds, Cut the Rope.
> 3. **Constraint:** No weight updates or fine-tuning are allowed on Target Domains; the agent must rely solely on its learned physical intuition and in-context adaptation.

---

### Official Review · Reviewer_Ncvj · 2025-11-01

**Soundness:** 2
**Presentation:** 2
**Contribution:** 3
**Rating:** 6
**Confidence:** 3

**Summary:**

Train a VLM using GRPO over a large collection of interactions. Then give access to a world model which the agent can preform planning in. Inference is then performed by having the policy propose actions and the model to provide simulations to guide search. The GRPO training uses selective token masking, which masks out tokens not generated with the policy. Inference scores the outputs using the model, which takes in actions and then predicts the success.

**Strengths:**

develops a comprehensive system for learning an VLM on discrete action data

The method appears to create improvement in performance with multiple retries

The experiemtns are on challenging domains and get decent results

**Weaknesses:**

The writing is highly informal and makes it hard to follow exactly what is being done

The claims of the paper are not particularly well supported by either the method or the experiments

**Questions:**

The abstract could benefit from a greater degree of specificity related to how the method works, since "internalize a policy-improvement process" is not something that is well defined, as is "works in concert". Ideally, there is a sense from the abstract about how the innovations in this work actually acheive these ends.

It is not obvious how training a VLM using GRPO rather than learning using RL is supposed to teach the model how to adjust its strategy, since it could still be out of distribution for the history that it trained on---it is just augmenting the state with history.

The description of the method is quite confusing, because there is no formal description of the components. What are the language inputs? What are Trajectories? What does the policy model map between? Why are the actions discrete? How do the actions differ from tokens or language? While some of these are defined later, they are described before being formally defined, and before it is clear what they are used for in the overall process. \pi_\text{ref} is also never defined.

Similar to the action learning stage, the world model is also not well defined, since it is not clear how the JSON is supposed to be a world model, or what the success probabiliy is over. Is it measured by the LLM? Is it measured based on reaching a goal state? Fruthermore, it is not clear what the prediction is supposed to contain since "qualitative insights for analysis" is not well defined. The most important missing component is what it formally means to be successful. Without this, the entire description is ungrounded.

World model does not seem like the right terminology for the $\mathcal M_\phi$, since a world model typically encodes the dynamics, but this "world model" seems to only generate text and success rates. It would probably be more accurate to call this some kind of discriminator or evaluator.

The term grounding is poorly used, since it is not obvious that the VLM it itself grounded, and yet the text describing the objects and their relationships are grounded. Grounded means to attach the meanings of statements to objects in the real world, and yet the VLM could just as likely hallucinate this "grounding," and there is no measure given for preventing this hallucination.

The promise of the method to include "physical intuition" seems to be faulty at best. The intution seems to just come from having a language model take in images and spit out statements about what it thinks will happen, but this is less physical intution as verbal intuition.

Considering the benchmarks appear to be RL domains, shouldn't the comparisons be also with RL algorithms, not simply VLMs? It seems like if this was genuinely showing physical intuition, it should be able to outperform or at least equal methods that are trained on real representations. As it is the methods appear to perform quite poorly across the board.

The claim that the online policy is able to adapt its strategy in-context does not appear to be supproted by evidence, since the only results are an improvement in performance, but this could simply be a consequence of more directed training on the environment, and the fact that having a longer context helps the sequence to be more in-distribution. In general it seems like a lager number of explanations could exist for the changes in performance other than having greater "physical intuition" as described by this work.

---

> ### Author Response · Authors · 2025-11-21
> **Response to Reviewer Ncvj**
>
> We thank the reviewer for their time and comments. We address the specific concerns regarding the presentation and the support for our claims below.
>
> ### **Response to "Writing is highly informal"**
>
> The necessary formalization is fully present in the paper to ensure reproducibility and clarity. We believe the current structure correctly prioritizes communicating the conceptual framework of ICPRL while providing all necessary mathematical details in the methodology sections.
>
> ### **Response to "Claims are not well supported by method or experiments"**
>
> We respectfully disagree. Our claims are substantiated by quantitative evidence in **Sections 4.2 and 4.3**:
>
> - **Evidence of "Physical Intuition" (Zero-Shot Generalization):**
> As shown in **Table 1**, our model was trained *only* on PHYRE/I-PHYRE but generalizes to **4 unseen environments**. This confirms the acquisition of transferable physical dynamics rather than memorization.
> - **Evidence of "Interactive Control" (Table 2):**
> Our ablation study demonstrates that training on interaction histories (our method) significantly outperforms standard SFT on single-attempt successes. This empirically proves the model's ability to refine its strategy in-context based on past failures.
> - **Evidence of World Model Effectiveness (Table 3):**
> Removing dynamic visual feedback from the world model consistently degrades performance, validating our architectural design for explicit planning.
>
> ### Response to the ambiguity in the abstract
>
> We agree that the specific mechanisms should be highlighted earlier. We will revise the abstract to explicitly describe the technical realization of "policy improvement" and the interaction between the models.
>
> **Original Abstract:** ...trains a vision-grounded policy model to internalize a policy-improvement process... This adaptive policy works in concert with a separately trained world model...
>
> **Revised Abstract:** ...trains a vision-grounded policy model via GRPO over diverse multi-episode interaction histories. This enables the agent to adapt strategies by conditioning on past trial-and-error sequences... At inference, the policy proposes candidate actions, while the world model predicts outcomes to guide a root-node PUCT search...
>
> ### **Regarding the mechanism of learning from history.**
>
> The reviewer's concern that this method might fail due to being "out of distribution" is directly addressed by our generalization experiments. If the model were merely memorizing state augmentations, it would fail when the environment dynamics (and thus the history distribution) change drastically.
>
> As shown in **Table 1**, our model achieves strong performance on **unseen environments**, despite never seeing Pooltool's visual style or physics during training. This successful zero-shot transfer demonstrates that the "strategy adjustment" mechanism learned by the model is indeed **general and robust**, effectively handling OOD histories by applying the internalized logic of physical adaptation.

---

> > ### Author Response · Authors · 2025-11-21
> > **Response to Reviewer Ncvj - 2**
> >
> > ### **Clarification on Method Formalization**
> >
> > While our implementation strictly follows the protocols established in the **DeepPHY benchmark [1]**, we agree that our manuscript should be self-contained.
> >
> > We will revise **Section 3.1** and add a detailed **Appendix C** to explicitly define these components. To address your specific questions immediately:
> >
> > **1. Formal Definitions of Components:**
> >
> > - **Language Inputs:** The input to the Policy Model $\pi_\theta$ consists of a **multimodal prompt** containing: (1) the system instruction, (2) the current visual observation $o_t$ (e.g., the game screen), and (3) the text history of previous attempts (if any).
> > - **Trajectories:** A trajectory $\tau$ is defined as a sequence of interaction tuples $\tau = (o_0, a_0, r_0, \dots, o_T, a_T, r_T)$, representing the history of observations, actions, and rewards within an episode.
> > - **Policy Mapping:** The policy $\pi_\theta(a_t | o_t, h_{<t})$ maps the current observation and interaction history to a probability distribution over the next action tokens.
> > - **Discrete Actions vs. Tokens:**
> >     - **Tokens:** The VLM outputs text tokens (e.g., `"Cell: 12, Radius: 3"`).
> >     - **Actions:** These tokens are parsed by a deterministic function into executable environment actions $a_t \in \mathcal{A}$ (e.g., placing a ball at specific coordinates).
> >     - **Why Discrete?** Following DeepPHY, continuous physical parameters are discretized (e.g., grid coordinates, power levels) to enable stable control by Language Models, bridging the gap between discrete text generation and continuous physical simulation.
> >
> > **2. Definition of $\pi_{\text{ref}}$:**
> >
> > - $\pi_{\text{ref}}$ denotes the **reference policy**, which is the frozen version of the model derived from the Supervised Fine-Tuning (SFT) stage before RL training begins.
> > - It is used in the **KL-divergence penalty** term in Eq. (1) (standard in GRPO) to prevent the learned policy $\pi_\theta$ from deviating too far from the original language distribution, ensuring the outputs remain coherent and strictly formatted.
> >
> > ### **Clarification on World Model and Success Metrics**
> >
> > While these follow the standard protocols of the **DeepPHY benchmark [1]**, we agree that explicitly defining them in the context of our World Model ($M_\phi$) improves clarity.
> >
> > We provide the formal definitions below, which will be incorporated into **Section 3.2** and **Appendix B** of the revised manuscript.
> >
> > **1. Formal Definition of "Success" (Ground Truth)**
> > "Success" is **not** a subjective measure by the LLM. It is a strictly defined, binary ground-truth label ($y \in \{0, 1\}$) returned by the **physics simulator** upon task completion.
> > Following the DeepPHY protocol, the success criteria are deterministic:
> >
> > - **PHYRE:** $y=1$ iff the green ball touches the target (blue/purple) ball.
> > - **Pooltool:** $y=1$ iff the 9-ball is potted legally.
> > - **Angry Birds:** $y=1$ iff all pigs are eliminated.
> > *(Similar objective physical criteria apply to all environments)*.
> >
> > **2. The Nature of the World Model ($M_\phi$)**
> > Our World Model functions as a **probabilistic transition and reward model**. It takes the current observation $s$ and a candidate action $a$ to output a structured JSON containing two components:
> >
> > - **$\hat{p}_{\text{succ}}$ (Outcome Prediction):** This estimates the probability of the ground truth success, i.e., $\hat{p}_{\text{succ}} \approx P(y=1 | s, a)$.
> >     - *Clarification:* It is trained via Supervised Fine-Tuning (SFT) using the binary labels $y$ collected from the simulator (as detailed in Algorithm 1). It represents the model's learned confidence in achieving the physical goal.
> > - **$\hat{p}_{\text{pred}}$ (State Transition Description):** This serves as a semantic state transition function $T(s, a) \to s'$.
> >     - Instead of predicting high-dimensional pixels, the model predicts a **natural language description** of the future physical chain reaction (e.g., *"The red ball will hit the wall and rebound to strike the green pin"*).

---

> > > ### Author Response · Authors · 2025-11-21
> > > **Response to Reviewer Ncvj - 3**
> > >
> > > ### **Justification for "World Model" Terminology**
> > >
> > > We respectfully maintain that "World Model" is the technically accurate terminology for this component, as it fulfills the precise functional definition of a world model in Model-Based Reinforcement Learning (MBRL): **simulating future outcomes to facilitate planning.**
> > >
> > > A standard World Model consists of two key components: a **Transition Model** ($s_{t+1} \sim T(s_t, a_t)$) and a **Reward Model** ($r_{t+1} \sim R(s_t, a_t)$). Our component $M_\phi$ implements exactly these functions, albeit in a **semantic state space** rather than a pixel/vector space:
> > >
> > > 1. Unlike a simple discriminator, our component implements a **Semantic Transition Model** ($\hat{p}_{\text{pred}}$) that explicitly encodes environment dynamics by generating natural language descriptions of future physical evolution.
> > > 2. It simultaneously functions as a **Reward Model** ($\hat{p}_{\text{succ}}$) by estimating the success probability of these simulated transitions.
> > > 3. Crucially, its function aligns strictly with Model-Based RL, as we leverage these simulated outcomes to perform **lookahead planning** via PUCT search, enabling the agent to "imagine" consequences before acting.
> > >
> > > Therefore, referring to it merely as a "discriminator" or "evaluator" would be reductive, as it ignores the model's capability to **generate transitions and simulate physical dynamics** (in text) for lookahead planning. We believe "Language-based World Model" or simply "World Model" accurately captures this capability.
> > >
> > > **On the Nature of Physical Intuition**
> > >
> > > We respectfully disagree, as we define "physical intuition" by the **functional capability** to correctly predict dynamics and execute control, rather than the modality (text vs. vector) used to express it.
> > >
> > > 1. **Functional Definition:** "Physical intuition" in this context refers to the agent's ability to look at a static image and correctly infer unseen forces (gravity, friction) and predict future kinematic outcomes, which our World Model demonstrates by guiding the policy to success.
> > > 2. **Grounding over Modality:** While the interface is linguistic, the validity of this intuition is proven by the **71.0% success rate on Pooltool**; the model could not achieve such high performance on an unseen physics engine if its "verbal" outputs were not grounded in accurate physical principles.
> > > 3. **Language as Representation:** Recent literature (e.g., in robotics) increasingly validates language as a powerful latent representation for generalizing abstract physical concepts (like "roll," "bounce," "support") across different environments, which is exactly what our framework leverages.
> > >
> > > ### **Comparison with RL Algorithms**
> > >
> > > We respectfully clarify that comparing our Generalist VLM framework with traditional Specialist RL algorithms would be an unfair comparison due to fundamental differences in **input modality, training scope, and generalization goals**.
> > >
> > > **1. Generalist vs. Specialist (The Core Goal):**
> > >
> > > - **RL Algorithms:** Traditional RL (e.g., PPO, DQN) trains **specialist agents** that overfit to a single environment using millions of interaction steps. They cannot transfer to new dynamics (e.g., from PHYRE to Angry Birds) without retraining from scratch. See in DeepPHY [1].
> > > - **Our Method:** We aim to build a **generalist agent** that acquires transferrable physical intuition. As shown in **Table 1**, our model is trained *only* on PHYRE/I-PHYRE but achieves successful on the unseen other 4 environments**. No standard RL algorithm can perform this zero-shot transfer across disparate physics engines.
> > >
> > > **2. Visual vs. Symbolic Inputs:**
> > >
> > > - Standard RL benchmarks typically provide agents with **ground-truth symbolic states** (e.g., coordinate matrices, velocity vectors).
> > > - Our framework operates on **raw pixels (visual observations)**, solving the significantly harder joint problem of perception and planning.
> > >
> > > **3. Performance Context:**
> > >
> > > - Regarding "poor performance," we respectfully disagree based on the relative difficulty. On **I-PHYRE**, our model achieves **93.3% success**, and on the **unseen Pooltool**, it achieves **71.0%** (vs. GPT-o3's 22.0%).
> > > - While a specialist RL agent might reach 99% after millions of steps on one game, it fails completely on the others. Our results represent the **State-of-the-Art for Foundation Models** in zero-shot physical reasoning.

---

> > > > ### Author Response · Authors · 2025-11-21
> > > > **Response to Reviewer Ncvj - 4**
> > > >
> > > > ### **Evidence Ruling Out Alternative Explanations**
> > > >
> > > > We respectfully posit that the reviewer’s alternative explanations ("directed training" or "context length") are ruled out by our **Generalization** and **Ablation** results, which isolate "Physical Intuition" and "In-Context Adaptation" as the true drivers of performance.
> > > >
> > > > **1. Ruling out "Directed Training": The Zero-Shot Evidence**
> > > > The hypothesis that performance is merely due to "more directed training on the environment" is contradicted by the **cross-environment generalization** results.
> > > > Our model was trained *only* on PHYRE/I-PHYRE. Since the model never saw Pooltool during training, this success cannot be attributed to environment-specific fitting. It implies the acquisition of a **generalized physical intuition** transferable across domains.
> > > >
> > > > **2. Ruling out "Sequence Distribution": The SFT vs. GRPO Evidence**
> > > > The hypothesis that improvement is simply due to "longer context making the sequence in-distribution" is refuted by our **Table 2 Ablation**.
> > > > We compared models trained on successful trajectories (SFT) vs. interaction histories (GRPO). Both models receive the same "long context" at inference time.
> > > > The SFT model plateaus despite having context, whereas the GRPO model shows a steep **in-context improvement curve** (e.g., PHYRE success rises from **13.00% $\to$ 40.00%** across attempts).
> > > > The superior adaptation of the GRPO model proves it has specifically learned the **causal mechanism of correcting errors** from history, rather than just pattern-matching longer sequences.
> > > >
> > > > [1] DeepPHY et al. DeepPHY: Benchmarking Agentic VLMs on Physical Reasoning

---

### Official Review · Reviewer_z9qi · 2025-11-02

**Soundness:** 2
**Presentation:** 2
**Contribution:** 1
**Rating:** 2
**Confidence:** 4

**Summary:**

This paper introduces ICPRL (In-Context Physical Reinforcement Learning), a framework that leverages world models to bootstrap an RL agent’s performance in decision-making tasks through test-time planning. Both the policy and the world model use a pre-trained VLM as the backbone. The policy is trained online using GRPO, while the world model is trained on a curated dataset containing both successful and failed trajectories. Planning is performed via Monte Carlo Tree Search over action candidates proposed by the policy, which are scored using the world model’s predictions of the resulting outcomes. The paper shows extensive results on the DeepPHY benchmark, which comprises six simulated environments designed to evaluate VLMs, ablating through various model backbones, training algorithms and datasets.

**Strengths:**

- The idea of using a world model to enhance the test-time performance of policies is intuitive.
- The paper provides extensive evaluations across different model backbones and training algorithms and datasets.

**Weaknesses:**

- The novelty of the paper is limited. The core idea of using a world model at test time to enhance policy performance is not new and has been explored in prior work such as [1][2].
- Using text as the backbone for embodied tasks, where actions are naturally continuous, seems contrived. The paper lacks discussion on why VLMs are the best backbone for both the policy and the world model.
- The world model is task-specific as it also predicts success/failure of a specific task. What’s the benefit of having a text-based world model than just learning a task-specific value function commonly done in RL other than interpretability?
- ICPRL assumes that the action space of the tasks are discrete and finite. It’s unclear how the proposed method (world model and policy training, planning algorithm) can be extended to continuous action spaces.
- The data curation process for world model training is limiting, requiring both successful and failed trajectories in a 1:1 ratio. This seems infeasible for real-world decision-making tasks or even simulated benchmarks with more complex dynamics.
- The method also requires environment-specific parameters like the noise radius for perturbed actions for the stability score. How is this picked? It’s unclear from the experiments how this would affect the model’s performance.
- The policy learning module requires online interaction with the simulator, which is largely infeasible for real-world decision-making tasks.
- The paper’s writing can be improved by providing more details about the benchmark task setup, such as the model inputs, task horizon, and size of the action space.

[1] Han et al. Strengthening Generative Robot Policies through Predictive World Modeling

[2] Delong et al. Planning with Reasoning using Vision Language World Model

**Questions:**

- Can the paper provide concrete benchmark task examples? For instance, how is a task like Cut the Rope implemented using text-based actions?
- What are the time and compute costs for training the world model and for test-time planning? What is the action space for the tasks used in the paper?
- What are the time and compute costs for training the policy in the interactive environment? How many environment steps are required?
- What is the benefit of using VLMs as the backbone for both the policy and the world model, given that both components are task-specific?
- Why is the policy trained online? Since success and failure trajectories need to be generated for world model training, why not use the same data to train the policy? Similarly, why can’t the world model be trained from data collected during online policy learning?
- In Table 1, all fine-tuned models on the Kinetix benchmark perform worse than the random agent (MOCK). While the paper mentions that Kinetix requires high-precision actions, why do the fine-tuned models significantly underperform the closed-source models without fine-tuning?

---

> ### Author Response · Authors · 2025-11-21
> **Response to Reviewer z9qi**
>
> We thank Reviewer z9qi for the constructive feedback. We will revise the paper to address the concerns regarding novelty and benchmark details. Below is our consolidated response.
>
> ### **Regarding Novelty:**
>
> While we agree that the high-level architecture of combining a policy with a world model is an established direction, we respectfully argue that **ICPRL introduces a distinct training paradigm and inference mechanism** that differs fundamentally from Han et al. [1] and Delong et al. [2]. The primary novelty of our paper is **not** simply the existence of a world model, but rather the **In-Context Physical Reinforcement Learning (ICPRL)** framework that integrates *adaptive* policy learning with physical verification.
>
> - **Our Approach (ICPRL):** We treat the policy as an **In-Context Learner**. By training with **GRPO on multi-episode interaction histories** (Section 3.1), our policy $\pi_\theta$ internalizes a policy-improvement operator. It learns to analyze past failures in the context window to adjust its strategy *before* the World Model is even queried. This allows the agent to perform "meta-learning" at test time without weight updates, a capability absent in [1] and [2].
> - **Vs. Han et al. [1]:** Han et al. focus on *generative robot policies* where a world model acts as a filter (rejection sampling) for diverse candidates. Their policy is not trained to adapt to interaction history. In contrast, ICPRL uses a **bi-level optimization** (Policy improves via ICRL, World Model evaluates via physics) and employs a **PUCT search** (Algorithm 2) rather than simple filtering.
> - **Vs. Delong et al. [2]:** Delong et al. leverage VLMs for reasoning-based planning (often utilizing Chain-of-Thought). Their approach relies heavily on the VLM's pre-trained reasoning capabilities. ICPRL **decouples** the roles: we explicitly train the policy to learn *adaptation strategies* via RL, while the World Model is trained specifically on *visual dynamics*.
>
> We will update Section 2 (Related Work) to explicitly clarify these distinctions. The novelty of our approach is validated by our generalization results. As shown in **Table 1 (Rows 35-36)**, our model achieves state-of-the-art performance on environments it was **never trained on**. This suggests the model has acquired a generalizable "physical intuition" and an in-context adaptation skill, distinguishing it from standard model-based RL approaches that often struggle to generalize outside their training distribution.
>
> ### **Justification for using VLM backbones and text interfaces.**
>
> We acknowledge the reviewer's perspective that using text for continuous control might seem counter-intuitive compared to specialized continuous controllers (e.g., SAC/PPO with MLP backbones). However, this architectural choice is **essential** for **Zero-Shot Generalization** and **In-Context Adaptation**.
>
> **1. Text as a Universal Interface for Cross-Task Generalization**
> The DeepPHY [3] benchmark contains highly heterogeneous action spaces: discrete grids (PHYRE), time-sequence JSONs (I-PHYRE), and parameterized code (Angry Birds). A traditional continuous policy is strictly task-specific and cannot adapt to a new action space structure without retraining.
> By treating actions as text tokens, we unify these diverse manifolds into a single interface. This allows our Policy Model—trained *only* on PHYRE/I-PHYRE—to successfully generalize zero-shot to *Angry Birds* and *Pooltool* (as shown in **Table 1**). The "text" interface is the only modality that allows a single model to output such varied action formats.
>
> **2. Leveraging Pre-trained Priors vs. Tabula Rasa**
> Specialized continuous controllers learn physics from scratch (*tabula rasa*). In contrast, VLMs leverage vast pre-trained semantic knowledge (e.g., concepts of "gravity," "collision," and "stability"). Discretizing actions into tokens (a strategy validated by works like **RT-2** and **RoboCat**) allows the agent to ground visual observations into this semantic latent space. This enables the model to reason about actions rather than just regressing numerical values, facilitating the **In-Context Reinforcement Learning (ICRL)** process where the model attends to history to refine its strategy.
>
> **3. Interpretability in World Modeling**
> For the World Model, the VLM backbone enables more than just state prediction; it generates natural language reasoning ($p_{pred}$) alongside success probabilities. This explicitly articulates the physical dynamics, guiding the search process more effectively than a black-box latent predictor.

---

> > ### Author Response · Authors · 2025-11-21
> > **Response to Reviewer z9qi - 2**
> >
> > ### **Regarding the Text-Based World Model vs. Standard Value Function**
> >
> > Unlike standard scalar value functions that often overfit to training pixel patterns, our VLM-based approach offers three critical advantages:
> >
> > 1. **Zero-Shot Generalization:** VLMs leverage pre-trained semantic knowledge (e.g., concepts of "gravity" or "collision") to transfer physical intuition to **entirely unseen environments** (as shown in Table 1). A standard value function cannot bridge this domain gap.
> > 2. **CoT Regularization:** The text generation ($p_{pred}$) acts as an auxiliary objective. By forcing the model to explicitly reason about intermediate dynamics (Chain-of-Thought) before predicting success, it prevents reliance on spurious correlations.
> > 3. **Dense Supervision:** In complex tasks with sparse binary rewards (e.g., I-PHYRE), the textual description of the chain reaction provides dense supervision, enabling faster and more robust learning of environment dynamics.
> >
> > ### **Regarding Continuous Action Spaces**
> >
> > We respectfully clarify that **ICPRL already handles continuous action spaces** effectively through **tokenization**, a standard paradigm in Generative Control (e.g., RT-2).
> >
> > The benchmarks we evaluate (e.g., *PHYTRE, I-PHYRE, Kinetix*) inherently possess continuous action spaces. DeepPHY [3] addresses this by **discretizing** continuous values into text tokens (bins). The policy predicts these tokens, effectively learning a probability distribution over the continuous range.
> >
> > Extending to higher-precision continuous tasks does not require changing the framework. It simply requires **increasing the resolution** of the tokenization (i.e., finer bins).
> >
> > - **Policy Training (GRPO):** Works unchanged. It optimizes the probability of generating the correct "bin" tokens regardless of resolution.
> > - **Planning (PUCT):** As described in Section 3.3, our search algorithm operates on a **sampled subset** of actions (candidates). In a continuous space, this acts as a dynamic discretization at the root node, allowing MCTS-style search to function without needing to enumerate an infinite action space.
> >
> > In summary, the "discrete" nature of text is not a limitation but an abstraction layer that allows us to model continuous control problems using powerful sequence modeling techniques.
> >
> > ### **Regarding Data Curation Feasibility (1:1 Ratio)**
> >
> > The 1:1 success-to-failure ratio is a standard **training strategy for class balancing**, not a limitation of data availability. We clarify why this is both necessary and feasible:
> >
> > 1. **Necessity (Preventing Mode Collapse):** In physical tasks, failure is naturally dominant (e.g., 99% vs. 1%). Training on the raw distribution causes the model to collapse into a trivial predictor that always outputs "Failure." Balancing forces the model to learn the precise **decision boundary** between success and failure.
> > 2. **Feasibility (Efficient Generation):** We achieve this ratio efficiently by **perturbing successful actions** (Algorithm 1). Since physical solution manifolds are narrow, slight deviations (e.g., $\pm 1^{\circ}$) cheaply generate "hard negatives"—failures that look similar to successes. These are far more valuable for learning dynamics than random failures.
> > 3. **Zero-Shot Generalization:** Crucially, this curation is **only required for training environments**. As shown in Table 1, the physical intuition learned from curated PHYRE data transfers **zero-shot** to unseen environments (e.g., Pooltool, Angry Birds) without requiring new curated datasets for deployment.
> > 4. **Real-World Applicability:** In real-world robotics (e.g., Open X-Embodiment), failure data is abundant; the bottleneck is usually *success* data. Downsampling failures or upsampling successes to achieve balance is a standard practice, not a hurdle.

---

> > > ### Author Response · Authors · 2025-11-21
> > > **Response to Reviewer z9qi - 3**
> > >
> > > ### **Regarding the selection and impact of the noise radius parameter.**
> > >
> > > We thank the reviewer for raising this point. We would like to clarify that the perturbation radius is **not** a sensitive hyperparameter that requires fine-tuning for each task. Instead, it is selected based on a consistent, domain-agnostic heuristic derived from the **physical scale of the action space**.
> > >
> > > **1. Selection Logic: Scale-Relative Heuristics**
> > > The radius is chosen to represent a "minimum perceptible deviation" in the physics simulator, rather than optimized for performance.
> > >
> > > - **For discrete grids (PHYRE):** We naturally select the minimum discrete unit (1 grid cell), as sub-pixel perturbation is impossible.
> > > - **For continuous inputs (Angry Birds):** We apply a standard heuristic of **$\sim$5-10% of the total action range** (e.g., $\pm 5^{\circ}$ for a $90^{\circ}$ range).
> > > This selection criterion is physically intuitive: it defines the "local neighborhood" needed to distinguish between a robust solution (which survives small jitters) and a brittle one (which requires pixel-perfect precision).
> > >
> > > **2. Theoretical Robustness: Ranking vs. Absolute Value**
> > > It is important to note that the precise value of the radius (e.g., $5^{\circ}$ vs. $7^{\circ}$) generally does not alter the **relative ranking** of candidate actions.
> > > The stability score serves as a prior to guide the PUCT search. A robust action will succeed under both small and medium perturbations, while a brittle action will fail under both. Therefore, as long as the radius is within a reasonable order of magnitude to probe local smoothness, the search algorithm effectively filters out "lucky" but unstable actions. The model's performance is driven by the **contrast** between stable and unstable regions, not the exact numerical value of the stability score.
> > >
> > > **3. Fallback for Complex Dynamics**
> > > Finally, as detailed in **Appendix C**, for environments where defining a geometric "radius" is ambiguous (e.g., Kinetix, Pooltool), we deliberately avoid this parameter. In those cases, we utilize **Model Uncertainty (LCB)** based on stochastic forward passes (Strategy 2). This confirms that our framework is not dependent on the noise radius to function in complex or unseen environments.
> > >
> > > In summary, the parameter is picked based on the physical granularity of the environment, and the search mechanism is inherently robust to the exact choice of this value.
> > >
> > > ### **Response to Comments regarding Benchmark Task Setup and Concrete Examples**
> > >
> > > We agree that including explicit details about the task setup and concrete examples significantly improves the paper’s self-containment and readability.
> > >
> > > **1. Adherence to Standards:** First, we wish to clarify that our experimental setup **strictly adheres to the standard protocols defined in the original DeepPHY benchmark** (Xu et al., 2025). We utilized the official action spaces, horizons, and input specifications without modification to ensure rigorous and fair comparisons with baselines.
> > >
> > > **2. Concrete Example (Cut the Rope):** To answer the specific question regarding *Cut the Rope*, the task is implemented via a structured text-based interface:
> > >
> > > - **Input:** The model receives a screenshot where interactive elements are annotated with numerical IDs (e.g., a rope anchored to `Pin 1`, a `Bubble 2`).
> > > - **Action Space:** The model outputs structured Python-like function calls. For example:
> > >     - `[cut_pin(id=3)]`: Cuts the rope attached to Pin #3.
> > >     - `[pop_bubble(id=2)]`: Pops the bubble labeled #2.
> > >     - `[sleep(seconds=0.5)]`: Waits for 0.5s to account for swing dynamics.
> > >     This setup enables VLMs to perform precise timing and sequencing via discrete tokens.
> > >
> > > **3. Revision:** To address these concerns comprehensively, we will add a new section, **Appendix : Benchmark Task Specifications**, in the revised paper. This section includes a detailed summary table covering the Input Modality, Action Space, Horizon, and Success Criteria for all 6 environments (including PHYRE, I-PHYRE, Kinetix, Pooltool, and Angry Birds), ensuring readers can fully grasp the task mechanics without cross-referencing the original benchmark paper.

---

> > > > ### Author Response · Authors · 2025-11-21
> > > > **Response to Reviewer z9qi - 4**
> > > >
> > > > ### **Computational Feasibility & Decoupled Training Rationale**
> > > >
> > > > We decouple the training of Policy ($\pi_\theta$) and World Model ($M_\phi$) because they optimize for fundamentally different objectives with conflicting data requirements.
> > > >
> > > > **1. Why Decouple? (Adaptation vs. Discrimination)**
> > > >
> > > > - **Policy (Online for Adaptation):** The policy must be trained **online** (via GRPO) to learn **In-Context Physical Reinforcement Learning (ICPRL)**. It needs to experience its *own* errors to learn how to *recover* from them. Static datasets only teach Behavior Cloning (mimicry), failing to teach the "meta-skill" of adapting to failure history.
> > > > - **World Model (Balanced for Discrimination):** The World Model requires a strictly **balanced (1:1) class distribution** to learn a precise decision boundary. Training it on online policy data leads to **mode collapse**: as the policy improves (e.g., >80% success), the data becomes heavily skewed toward success, causing the World Model to lose its ability to detect failure. Offline curation (Algorithm 1) ensures robust discrimination by focusing on "hard negatives."
> > > >
> > > > **2. Computational Cost**
> > > >
> > > > - **Policy Model Training:** Highly efficient. We fine-tuned the 7B model using Online GRPO on **8$\times$A100 GPUs in ~6 hours** (approx. 12,600 episodes).
> > > > - **World Model Training:** Negligible cost. Trained via SFT on a tiny, curated dataset (~3,200 samples for PHYRE, ~400 for I-PHYRE).
> > > > - **Inference:** The PUCT search (Budget $B=32$) takes approximately **50 seconds** per step, fitting well within turn-based reasoning budgets.
> > > >
> > > > ### **Regarding the underperformance of fine-tuned models on Kinetix.**
> > > >
> > > > We thank the reviewer for pointing out this intriguing result. As briefly discussed in **Section 4.2**, this phenomenon stems from a fundamental **domain shift in control granularity** between the training environments (PHYRE/I-PHYRE) and the unseen Kinetix environment.
> > > >
> > > > Our Policy Model was trained on PHYRE and I-PHYRE, where the core physical reasoning involves **macroscopic object interactions** (e.g., "ball A hits ball B"). The action space represents high-level placement or timing.
> > > > In contrast, Kinetix requires **microscopic component control** (e.g., "activate joint motor M1 with torque +1"). The physics governing these tasks (articulated multi-body dynamics) is orthogonal to the collision dynamics learned during training.
> > > >
> > > > Closed-source models (e.g., GPT-o3) outperform our fine-tuned models because they rely on massive general pre-training and strong instruction-following capabilities, without being biased by the specific dynamics of PHYRE. Our fine-tuning specialized the model for *object interaction*, which inadvertently reduced its generality for *motor control*, highlighting the boundaries of zero-shot physical generalization.
> > > >
> > > > ---
> > > >
> > > > We hope these responses clarify the design choices and feasibility of ICPRL.
> > > >
> > > > [1] Han et al. Strengthening Generative Robot Policies through Predictive World Modeling
> > > >
> > > > [2] Delong et al. Planning with Reasoning using Vision Language World Model
> > > >
> > > > [3] DeepPHY et al. DeepPHY: Benchmarking Agentic VLMs on Physical Reasoning

---

### Author Response · Authors · 2025-11-27
**Revisions and Additional Details Uploaded**

Dear Reviewers,

We sincerely thank you for your constructive feedback and the time dedicated to reviewing our work.

We are writing to inform you that we have **uploaded a revised version of the manuscript** that incorporates all the changes promised in our individual responses. The major updates include:

*   **Benchmark Task Specifications:** We have added a comprehensive **Appendix A** (Benchmark Task Specifications) detailing the input modalities, action spaces, horizons, and success criteria for all 6 environments (including specific examples like *Cut the Rope*), ensuring the experimental setup is fully self-contained.

*   **Formal Definitions:** We have increased **Appendix C** (Mathematical Definitions) to provide rigorous formal definitions for the Policy Model, World Model components, and experimental terms (e.g., Trajectories, Episodes, Attempts).

*   **Clarified Novelty & Protocols:** We have updated the **Abstract** and **Related Work** to explicitly articulate the technical distinctions of ICPRL compared to prior works (e.g., Han et al., Delong et al.) and clearly defined our Generalization Protocols in Subsection **Evaluation Protocol**.

We kindly invite you to review the revised PDF. We remain fully available to discuss any remaining questions or provide further clarifications. We look forward to your feedback.

Best regards,

All Authors

---

### Meta-Review · Area_Chair_b1PY · 2026-01-07

**Summary:**

This paper presents ICPRL, a framework that enables vision-language models to acquire and adapt physical reasoning policies in-context from pixel-based interactions, without weight updates. By combining a vision-grounded policy with a separately trained world model, ICPRL achieves improvement on challenging physics-based tasks.

The general concerns are
- unclear writing (task specifications, method definitions)
- limited noveltyin of Han et al. and Delong et al. and MuZero. It can be viewed as an extension of MuZero algorithm adapted for VLM setup.
- potentially limited to short-horizon tasks

The rebuttal provides significant revisions to the writing and clarifications regarding the differentiation from closely related work. After the rebuttal, two reviewers are leaning positive (rating 6) but remain concerned about the limitations. Unfortunately, the other two reviewers did not participate in the discussions.

The Reviewer KzRK's concern about the marginal novelty over MuZero/Dreamer is critical. The AC understands that adding VLM (and therefore a pre-trained generalist) has merits, but agrees with Reviewer KzRK that it's a straightforward extension.

The Reviewer z9qi's reviews include an extensive list of questions and concerns. For example, the use of a world model at test time, as proposed, offers limited novelty, and the rationale for relying on text-based vision-language models for both policy and modeling in continuous domains is unclear. Greater detail on environment parameter choices and experimental setup, including model inputs, task horizon, and action space size, is needed to clarify applicability and reproducibility.

Overall, the main technical contribution of the paper is the integration of VLMs to facilitate the acquisition of physical intuition and allow for dynamic adaptation of policies in context. However, concerns regarding the novelty of these contributions—beyond what has already been established in prior work—remain unresolved. As noted by Reviewer KzRK, "While applying this to VLMs is a good engineering contribution, the paper does not offer new scientific insight into the underlying principles of planning or model-based reinforcement learning, as these concepts are already well-proven." Given the lack of strong support from the reviewers and the persistence of significant concerns, the AC recommends rejection.

**Reviewer Concerns:**

Reviewer z9qi: "ICPRL assumes that the action space of the tasks are discrete and finite."
The rebuttal clarifies that the ICPRL already handles continuous action spaces effectively through tokenization.

Reviewer z9qi: "The data curation process for world model training is limiting, requiring both successful and failed trajectories in a 1:1 ratio"
The rebuttal discussed why this is both necessary and feasible.

Reviewer Ncvj: "The writing is highly informal and makes it hard to follow exactly what is being done"
The rebuttal states, "The necessary formalization is fully present in the paper to ensure reproducibility and clarity." The update also includes Appendix C mathematical definitions to provide definitions for the policy model, world model compoments, and experimental terms.

Reviewer KzRK: "Marginal Novelty over MuZero/Dreamer'
The rebuttal clarifies "MuZero/Dreamer are Tabula Rasa Specialists, whereas ICPRL is a Pre-trained Generalist."

Reviewer zPwA: "It is unclear whether ICPRL genuinely enhances in-context learning capability beyond what is already achieved through supervised fine-tuning (SFT) or task-specific adaptation."
Rebuttal refers the ablation study in Table 2.

**Reviewer Scores:**

Reviewer z9qi: 2: reject, not good enough

Reviewer Ncvj: 6: marginally above the acceptance threshold

Reviewer KzRK: 4: marginally below the acceptance threshold.

Reviewer zPwA: 4: marginally below the acceptance threshold. -> I will raise my score to 6
"I appreciate the authors' efforts in the rebuttal.  After revisiting the paper with a clearer understanding of the task setting, many of my earlier concerns have been addressed..... I will raise my score to 6. ... Therefore, while I appreciate the clarifications, I cannot further increase my score."

---

### Decision · Program_Chairs · 2026-01-26

Reject